# Mitochondria localization induced self-assembly of peptide amphiphiles for cellular dysfunction

M.T. Jeena[1], L. Palanikumar[1], Eun Min Go[2], Inhye Kim[3], Myoung Gyun Kang[1], Seonik Lee[4], Sooham Park[1], Huyeon Choi[1], Chaekyu Kim [1], Seon-Mi Jin[3], Sung Chul Bae [4], Hyun Woo Rhee[1], Eunji Lee[3], Sang Kyu Kwak [2] & Ja-Hyoung Ryu[1]

Achieving spatiotemporal control of molecular self-assembly associated with actuation of biological functions inside living cells remains a challenge owing to the complexity of the cellular environments and the lack of characterization tools. We present, for the first time, the organelle-localized self-assembly of a peptide amphiphile as a powerful strategy for controlling cellular fate. A phenylalanine dipeptide (FF) with a mitochondria-targeting moiety, triphenyl phosphonium (Mito-FF), preferentially accumulates inside mitochondria and reaches the critical aggregation concentration to form a fibrous nanostructure, which is monitored by confocal laser scanning microscopy and transmission electron microscopy. The Mito-FF fibrils induce mitochondrial dysfunction via membrane disruption to cause apoptosis. The organelle-specific supramolecular system provides a new opportunity for therapeutics and in-depth investigations of cellular functions.

[1] Department of Chemistry, Ulsan National Institute of Science and Technology (UNIST), Ulsan 44919, Republic of Korea. [2] School of Energy and Chemical Engineering, Ulsan National Institute of Science and Technology (UNIST), Ulsan 44919, Republic of Korea. [3] Graduate School of Analytical Science and Technology, Chungnam National University, Daejeon 34134, Republic of Korea. [4] Department of Biological Engineering, Ulsan National Institute of Science and Technology (UNIST), Ulsan 44919, Republic of Korea. Correspondence and requests for materials should be addressed to E.L. (email: eunjilee@cnu.ac.kr) or to S.K.K. (email: skkwak@unist.ac.kr) or to J.-H.R. (email: jhryu@unist.ac.kr)

Molecular self-assembly is one of the most attractive strategies for creating functional materials for integral biomedical applications such as drug/gene delivery and bio-sensing[1, 2]. Nanostructural assemblies of amphiphilic peptides and their biological functions have been extensively investigated in recent decades[3, 4]. In particular, in-situ assembly of such building units with accompanying cellular functions inside a living cell (i.e., intracellular assembly) and their interaction with cellular components have been emerging as a versatile strategy in controlling cellular fate[5–7]. However, achieving spatiotemporal control (i.e., inside cellular organelles or other sub-compartments) over the self-assembly of synthetic molecules inside the cell is challenging because of the difficulty of studying their behavior in the complex intracellular environment.

Self-assembly is an equilibrium process between the individual building units and their aggregated state, and the concentration of the molecules should be over the critical value to induce assembly (i.e., the critical aggregation concentration (CAC)). In living cells, achieving the CAC is also required to form assemblies of individual molecules, but has a limitation because the chemical complexity of cellular environments disrupts interactions among synthetic building units. Intracellular self-assembly thus requires a higher concentration of the molecules than the CAC, which may limit the practical implementations of self-assembling molecules. Transformation of the molecular structure from hydrophilic to hydrophobic units inside the cell (or pericellular space) through external stimuli (chemical, or physical) is a powerful strategy to reduce the CAC by increasing the propensity for self-assembly. However, chemical and physical stimuli (e.g., light, temperature, pH, and redox) are not relevant for intracellular assembly because they induce severe damage to cell. Recently, enzyme instructed intracellular self-assembly (EISA) has emerged as an effective approach to induce cellular dysfunction, in which an enzyme changes the molecular structure of precursor from soluble hydrophilic to self-assembling units to form ordered-structure within the cell or pericellular space[8, 9]. However, EISA could not be generalized to various cancer cells, since the usage of biological stimuli (e.g., enzymes) is usually limited to specific cell types or cellular compartments. Moreover, the EISA occurs via an enzymatic conversion (chemical reaction involving bond-breaking) inside the cell or near the cell surface, which is a time-consuming process.

We here hypothesize that a specific cellular organelle-localization induced supramolecular self-assembly (OLISA) system could be a general strategy to induce self-assembly by raising local concentrations of the self-assembling molecules without additional treatment. The small molecules readily diffuse through the cell membrane, reach to the target site (organelle or sub-cellular compartment depending on the targeting moiety), and then they undergo self-assembly inside the targeted organelle as a result of increased local concentration. The accumulation of molecules inside an organelle like mitochondria is ~500–1,000 times higher than that of extracellular space[10]. While EISA which is usually happened inside the cell or near the cell surface requires very high concentration of molecules (over several hundreds micromole), OLISA occurs with low dosage concentration (several tens micromole), which is a superior advantage of OLISA. The importance of such a strategy is that the activity is turned on once the assembly is induced inside the organelle. We designed mitochondria-accumulating amphiphilic peptide (Mito-FF), which consists of diphenylalanine as a β-sheet-forming building block, TPP as a mitochondrial targeting moiety, and pyrene as a fluorescent probe. Mito-FF favorably accumulated in the mitochondria of cancer cells because of the high negative membrane potential and the increased concentration caused Mito-FF to self-assemble into a fibrous structure, whereas lack of fibril formation was observed in normal cells[10, 11]. The stiff Mito-FF fibrils destroyed the mitochondrial membrane and activated the intrinsic apoptotic pathway against cancer cells (Fig. 1). This OLISA system offers a new approach for targeted cancer chemotherapy.

## Results
**Self-assembly and mitochondrial accumulation of Mito-FF.** The Mito-FF molecular design consists of Phe-Phe-Lys (FFK) tripeptide as a backbone, synthesized via standard solid-phase synthesis (SSPS) (Supplementary Figs. 1 and 2). The N-terminus

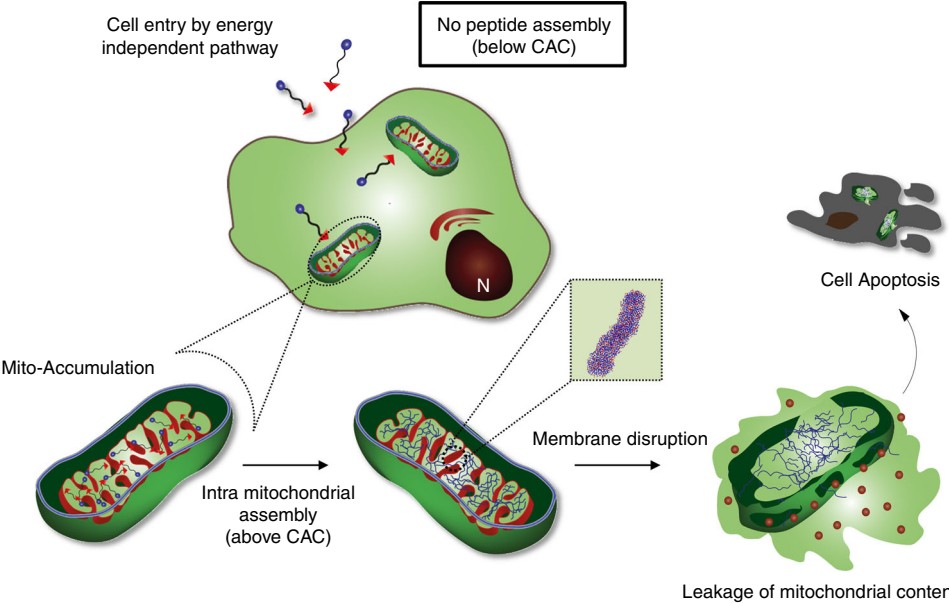

**Fig. 1** Intra-mitochondrial assembly of Mito-FF. The self-assembly process is driven by the increased mitochondrial membrane potential of cancer cells, leading to high mitochondrial accumulation of Mito-FF followed by self-assembly into fibrils. The intra mitochondrial fibrils further disrupt the membrane, resulting in leakage of mitochondrial contents to the cytosol, which eventually induces cellular apoptosis

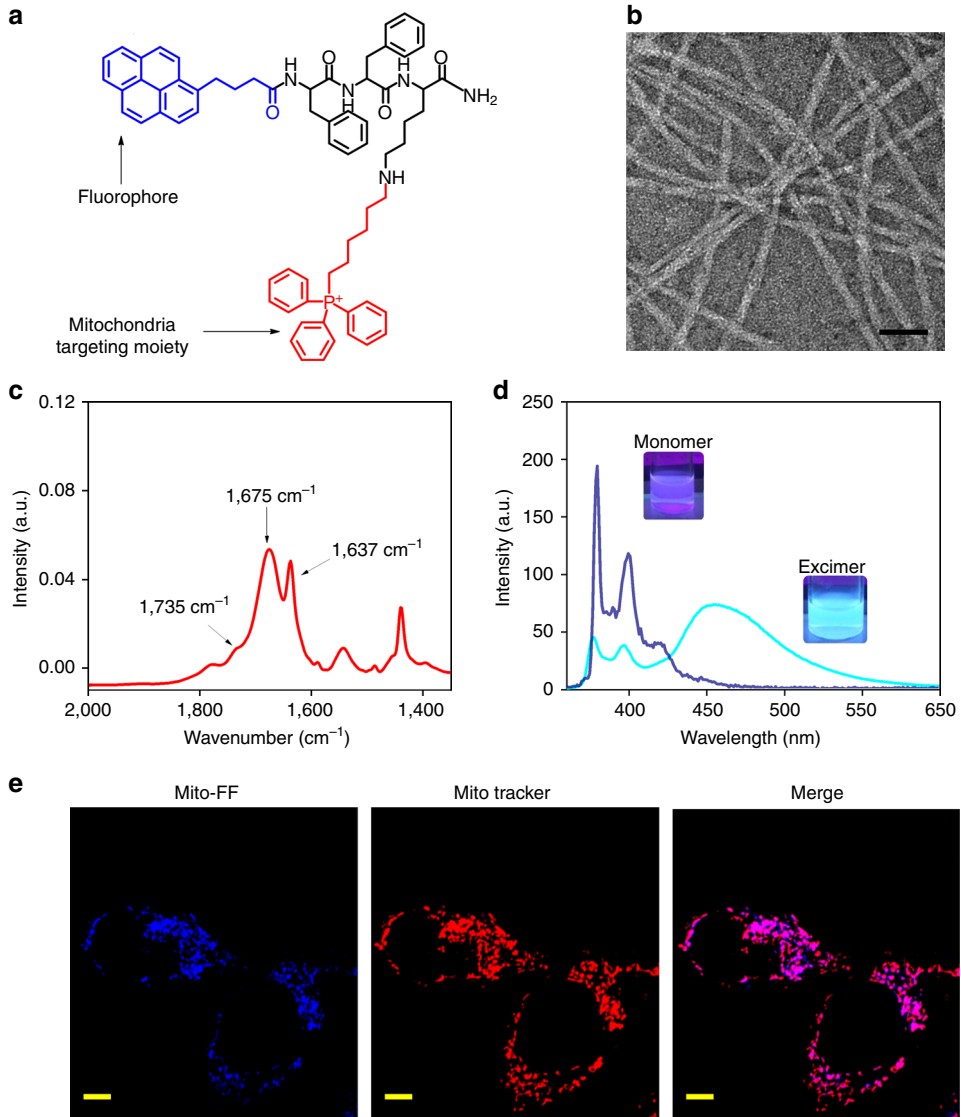

**Fig. 2** Self-assembly and mitochondrial localization of Mito-FF. **a** Structural design of the mitochondria-targeting peptide amphiphile, Mito-FF, which is equipped with a pyrene group for fluorescence detection inside cells and TPP for targeting of mitochondria. **b** TEM image of Mito-FF fibril in water (scale bar, 50 nm). **c** FT-IR spectra of Mito-FF fibril showing the β sheet confirmation. **d** Emission spectra for Mito-FF showing the excimeric emission above the CAC (*sky blue line*) and lack of excimer below the CAC (*deep blue line*) (the inset shows optical images of monomeric (*left*) and excimeric (*right*) emission). **e** Mitochondrial co-localization of Mito-FF measured with MitoTracker Red FM shows high localization inside mitochondria (scale bar, 5 μm)

of FFK was conjugated with pyrene butyric acid, which not only serves as a fluorophore but also increases the propensity for self-assembly by enhancing hydrophobic and π–π interactions, and the amine group of the lysine side chain was conjugated to TPP for targeting of mitochondria (Fig. 2a). Mito-FF self-assembled into nanofibers with a width of $9.6 \pm 1.1$ nm (mean ± s.d.) averaged over 100 nanofibers (in triplicate) for the individual fibers and a length of several micrometers in phosphate-buffered saline (PBS) (Fig. 2b and Supplementary Fig. 3). The CAC was determined to be ~60 μM by examining the excitation spectra of pyrene, further confirmed by performing surface tension analysis (Supplementary Figs. 4 and 5). Fourier transform-infrared analysis (FT-IR) indicated an amide I band near 1,637 cm$^{-1}$, indicating the presence of a β-sheet structure confirmation (Fig. 2c). Spectrofluorometric analysis exhibited intense sky blue emission at 450 nm in PBS above the CAC, indicative of pyrene excimer formation in the aggregation state. However, the emission was significantly reduced to a pale blue color in methanol and in PBS below the CAC, giving emission in the range of

370–410 nm reflective of pyrene in its non-aggregated state (Fig. 2d). Confocal microscopy showed a good overlap between the red fluorescence of MitoTracker and the blue fluorescence of Mito-FF (Pearson's Coefficient (Rr) was calculated as + 0.8)[12] (Supplementary Fig. 6). Cellular uptake analysis at 4 °C showed similar uptake as that of 37 °C suggesting that the peptide enters via an energy-independent pathway (Supplementary Fig. 7). Although the assembled structure and high-molecular-weight molecules generally are not capable of cellular uptake and require a complex energy-dependent mechanism such as endocytosis, Mito-FF overcomes this barrier because it diffuses through the plasma membrane and is readily internalized.

**Intra-mitochondrial assembly**. The mitochondrial accumulation of Mito-FF was analyzed to determine whether the effective concentration inside the mitochondria could exceed the CAC required for self-assembly (Supplementary Fig. 8). Mito-FF was incubated with HeLa cells for 3 h and its uptake was measured by

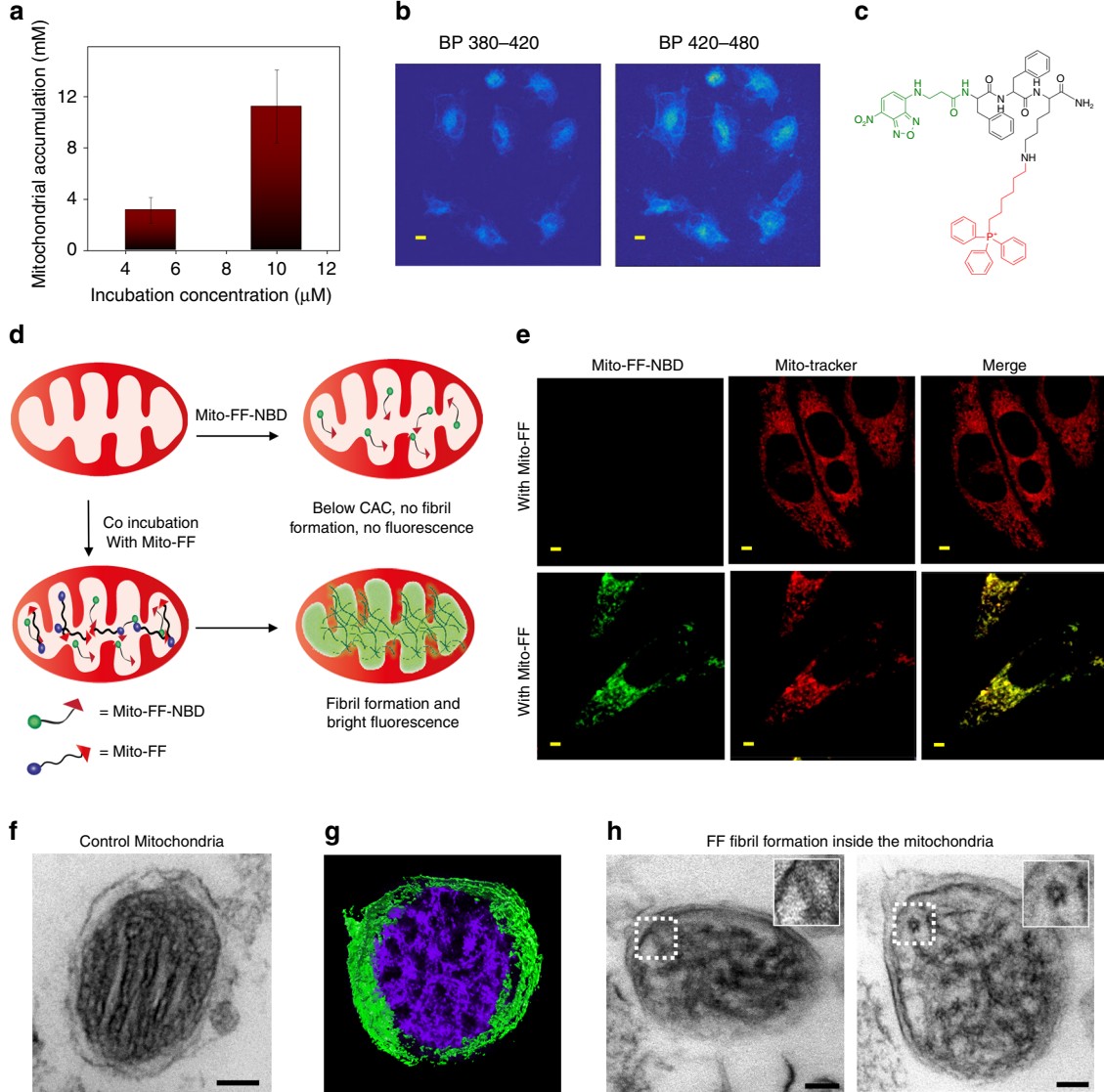

**Fig. 3** Intra-mitochondrial assembly. **a** Mitochondrial accumulation of Mito-FF in HeLa cells. Data represent mean ± s.d. from three independent experiments. **b** Two-photon analysis of Mito-FF-treated HeLa cells showing high-intensity excimer emission with a 420–480 nm filter, compared with a 380–420 nm filter (which corresponds to monomeric emission) (scale bar, 5 μm). **c** Molecular structure of Mito-FF-NBD. **d** Schematic diagram showing co-assembly of Mito-FF-NBD with Mito-FF. **e** Co-assembly inside mitochondria indicated by the bright green fluorescence of Mito-FF-NBD in presence of Mito-FF (*lower panel*); however, such emission was not observed with Mito-FF-NBD alone (*upper panel*) (scale bar, 2 μm). **f** TEM image for isolated mitochondria (control). **g** 3D volume image of Mito-FF fibrils inside mitochondria isolated from a C3H female nude mouse brain, and **h** construction images showing the side and cross-sectional views of fibrils at different tilt-angles of −60° and +20° (*from left to right*) (scale bar, 100 nm)

monitoring the emission spectra of Mito-FF in the cell lysate[13] (Supplementary Fig. 9). Our calculations showed that the mitochondrial accumulation of Mito-FF increased in accordance with the original concentration in the culture medium, i.e., 3.1 ± 1 and 11.2 ± 2.6 mM for external concentrations of 5 or 10 μM, respectively, suggesting that the self-assembly of Mito-FF inside the mitochondria could readily occur as a result of the high local concentration (Fig. 3a). We envisioned that intracellular self-assembly of Mito-FF would result in excimeric emission from the pyrene, similar to the behavior of Mito-FF in the bulk solution (Fig. 2d). To study the excimeric emission of Mito-FF in HeLa cells, we performed a two-photon microscopic analysis of HeLa cells pretreated with Mito-FF at a two-photon excitation wavelength of 780 nm (Fig. 3b). The emission analysis through different band-pass (BP) filters for wavelengths 420–480 (for detection of excimeric emission) and 380–420 nm (for detection of monomeric emission) showed different intensities of emission.

The 420–480 nm filter provided much higher-intensity emission, which indicates that the Mito-FF fibrils formed inside mitochondria, giving rise to excimer emission from the pyrene in the aggregated state.

To further study the fibril formation inside the mitochondria, another designer peptide was synthesized: Mito-FF-NBD, where pyrene was replaced with the environmentally sensitive dye, 4-nitro-2, 1, 3-benzoxadiazole (NBD) (Fig. 3c). NBD is highly fluorescent in hydrophobic environments and has been used for imaging biological components[14]. Recently, Xu et al. reported that NBD could serve as an efficient reporter for intracellular fiber formation with peptide-based small molecules based on its bright fluorescence upon intracellular fiber formation[15]. The co-assembly of Mito-FF-NBD with Mito-FF into fiber assembly in PBS was confirmed by molecular simulation (CGMD) (Supplementary Fig. 10) and TEM analysis (Supplementary Fig. 11). The TEM showed that the co-assembled fibers have a

decreased diameter ($7.1 \pm 0.8$ nm averaged over 100 nanofibers in triplicate) compared with Mito-FF nanofibrils alone or Mito-FF-NBD alone, suggesting that they assemble each other. We co-incubated HeLa cells with Mito-FF-NBD (10 μM) and Mito-FF (10 μM) and analyzed fluorescence emission using confocal microscopy. Only Mito-FF-NBD showed diminished fluorescence. In contrast, co-incubation of Mito-FF-NBD and Mito-FF resulted in intense green emission of NBD and good co-localization with MitoTracker (Fig. 3d, e). Mito-FF-NBD could form fibrils above their CAC as shown in Supplementary Fig. 12. We observed that Mito-FF-NBD showed a concentration dependent fluorescence inside the cell. No fluorescence observed for 10 μM dosage concentration, negligible fluorescence for 20 μM and bright green fluorescence starts appearing above 40 μM, which well-overlaps with red fluorescence of MitoTracker as indicated by the 2D and three dimensional (3D) confocal images (Supplementary Figs. 13 and 14). It indicates that much higher concentration is needed for Mito-FF-NBD to rise intra-mitochondrial concentration above the CAC compared with Mito-FF. When we have analyzed the IMC of Mito-FF-NBD in the HeLa cells by using the same procedure as described for Mito-FF, we found very low cellular uptake for 10 μM while the IMC increased to $3.2 \pm 0.7$ mM at high dosage concentration (100 μM). These results suggest that Mito-FF-NBD could co-assemble well with Mito-FF and NBD located inside the hydrophobic fiber to emit bright green fluorescence. However, Mito-FF-NBD without Mito-FF could not self-assemble into the fiber because it could not reach to the high CAC of 1.5 mM in the mitochondria with an external concentration of 10 μM.

To visualize intra-mitochondrial self-assembly, TEM images were taken after application of Mito-FF to mitochondria isolated from a mouse brain (C3H female)[16]. The TEM images showed fibrillar structures with a diameter of $9.0 \pm 1.5$ nm averaged over 100 nanofibers (triplicate) within the mitochondria (Supplementary Fig. 16). For clear demonstration of the nanofibers within the mitochondria, TEM tomography (TEMT) was performed. TEMT is a powerful method to study the internal architecture of the cellular structure or macromolecular complexes[17]. In here, depending upon the 3D orientation of the fibrils within the mitochondria and internal membrane at different tilt angles, we could differentaite the fibrils which are cylindrial in shape from the internal mitochondrial membranes which have a lamellar structure. Using TEMT, we figure out the fibrils within the mitochondria with respect to tilt angles from $-68°$ to $+68°$, where at $-60°$ focused fibril has observed in its surface view and at $+20°$ the same fibril has observed from the top as shown in Fig. 3h ($-60°$ (left), $+20°$ (right)). The mitochondrial cristae enclose a dense protein-rich matrix and are surrounded by an energy-transducing inner membrane and an outer membrane. As shown in Fig. 3g, the fibril formation altered the topology of the mitochondrial membrane compared with the untreated control (Fig. 3f). As a consequence, a considerable adverse effect on cell function may be expected, as the mitochondria play a vital role in controlling cellular metabolism.

**Mito-FF fibrils induce mitochondrial dysfunction.** The mitochondria showed severe morphological damage with fragmentation after 1 h incubation of Mito-FF in HeLa cells (Fig. 4a). To further investigate the mitochondrial dysfunction induced by Mito-FF fibrils, the membrane potential depolarization of the mitochondria was analyzed by using tetramethyl rhodamine dye (TMRM), which shows bright fluorescence under normal conditions that vanishes following membrane depolarization[18]. Confocal analysis of TMRM-labeled HeLa cells after treatment with Mito-FF showed robust red fluorescence after a 1 h

incubation, which began to diminish within 3 h and had completely vanished after 6 h (Fig. 4b), suggesting that the Mito-FF fibrils in the mitochondria adversely affected their function. This could imbalance mitochondrial reactive oxygen species (ROS) production; excess production of ROS contributes to mitochondrial damage in several pathologies and causes mitochondrial DNA damage, membrane disruption, and protein damage. The excess mitochondrial ROS production induced by Mito-FF fibrils was monitored with MitoSOX Red, which showed red fluorescence after 6 h of incubation with Mito-FF (Supplementary Fig. 17). Furthermore, staining with dihydroethidium dye (DHE), which intercalates into nuclear DNA and oxidizes to the ethidium ion to show red fluorescence in the presence of ROS, showed strong red emission in HeLa cells after treatment with Mito-FF, suggesting that the Mito-FF fibrils markedly induced cell damage[19] (Fig. 4c).

Diphenylalanine is a crucial building block for accelerating the amyloid assembly process in Alzheimer's disease and other neurodegenerative diseases, and it ultimately induces toxicity by the formation of toxic fibril aggregates[20]. We speculate that the mitochondrial damage arose specifically from the fibril formation of diphenylalanine rather than other morphologies. To investigate this further, our molecular design was extended to Mito-VV and Mito-$F_xF_x$, where phenylalanine was replaced by valine and cyclohexylalanine, respectively (Fig. 4d, e). Both peptides assembled into fibrous structures with a CAC of 70 and 30 μM, respectively (Fig. 4g, j and Supplementary Fig. 18). The membrane depolarization ability of in HeLa cells was evaluated, and both peptides could efficiently depolarize the mitochondrial membrane within 6 h, which was followed by ROS generation as evidenced with DHE and MitoSOX Red (Fig. 4i and Supplementary Fig. 19). Notably, Mito-GG, a designer peptide forming a micelle (~50 nm size above the CAC of 114 μM) (Supplementary Fig. 20) showed no membrane depolarization by TMRM (Fig. 4n). Additionally Mito-GG induced considerably less ROS production as shown in Fig. 4o. From these results, we considered that mitochondrial stress induced by the small micelles might be significantly lower than that induced the fibril analogs, resulting in a reduced extent of dysfunction caused by Mito-GG.

A TEM experiment for HeLa cells treated with Mito-FF (20 μM for 3 h) was conducted to investigate the structural change of mitochondria induced by intra-mitochondrial fibril formation[21]. As shown in Fig. 4p, the mitochondria were found to be severely damaged with distorted membrane after Mito-FF treatment, and there were devoid of normal mitochondria morphology within HeLa cells. It was observed that the fibrils were centered within the destroyed mitochondria. It might be expected that fibril formation inside the mitochondria induces damage to the mitochondria with membrane disruption. However, the mitochondria in HeLa cells treated with Mito-GG remained unaffected (Fig. 4q) under similar conditions and appeared identical to the control (Fig. 4r). These results indicate that fibril assembly plays an important role in mitochondria dysfunction by disruption of the mitochondrial membrane, whereas spherical assembly does not.

**Proposed mechanism for mitochondrial dysfunction.** We speculate that the mitochondrial dysfunction induced by the supramolecular assembly resulted from a loss of mitochondrial membrane integrity and consequent leakage of mitochondrial content. Our analyzes above suggest that a fibrous assembly, rather than a spherical assembly, was primarily responsible for the loss of mitochondrial membrane integrity (Fig. 4). The high positive surface charge ($+43 \pm 1.3$ mV (triplicate measurement),

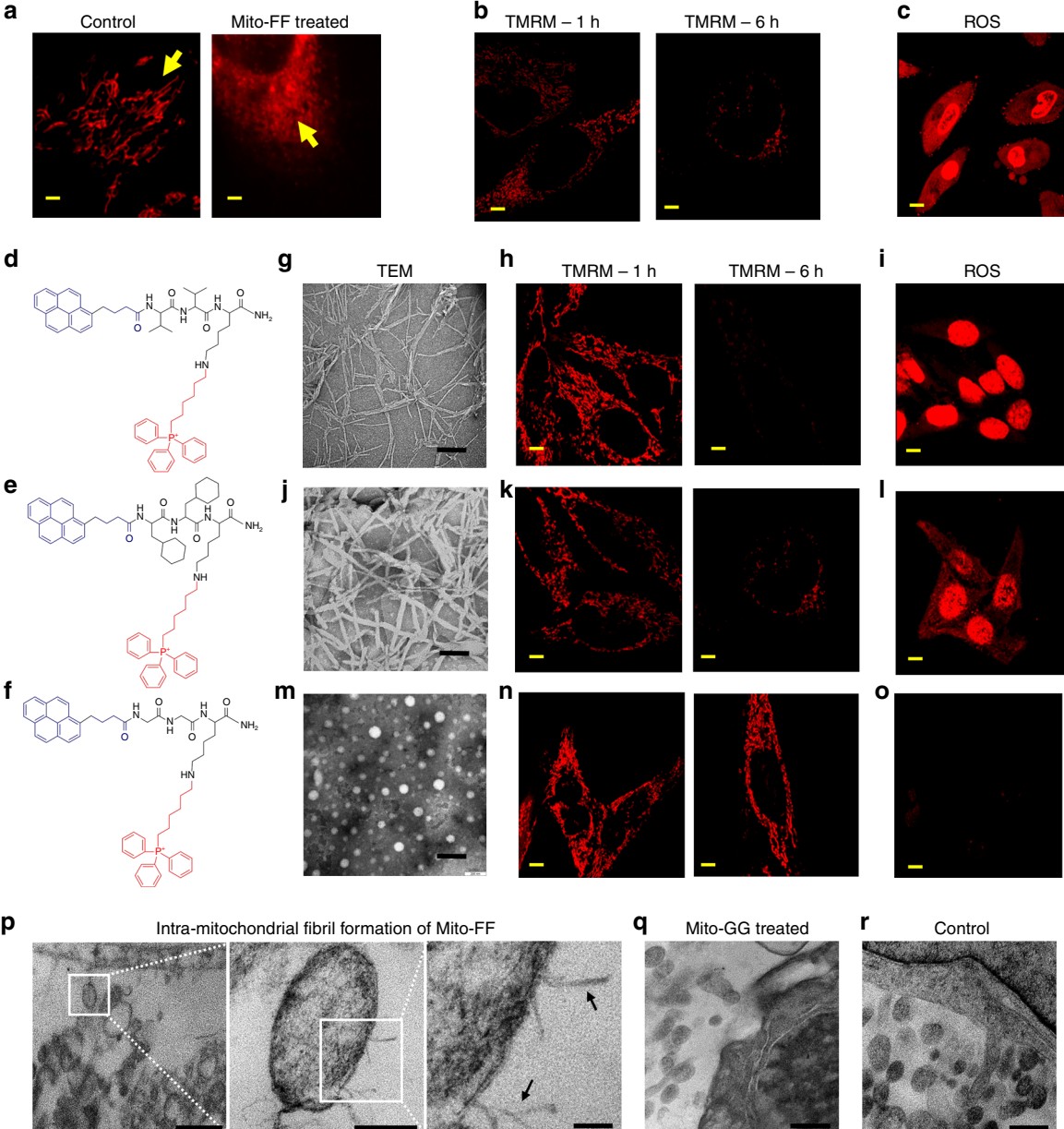

**Fig. 4** Mitochondrial dysfunction. **a** Confocal analysis showing mitochondrial shrinkage after Mito-FF treatment on HeLa cells (*right*), compared with control (*left*) (scale bar, 1 μm). **b** Membrane depolarization of mitochondria over time after treatment with Mito-FF was measured by TMRM (scale bar, 2 μm). **c** ROS generation measured by DHE (scale bar, 5 μm). **d–f** Molecular design of the control peptides. **g** TEM image of Mito-VV (scale bar, 500 nm). **h** Membrane depolarization induced by Mito-VV after 1 or 6 h (scale bar, 2 μm), and **i** intracellular ROS generation (scale bar, 5 μm). **j** *From left*: TEM image of Mito-$F_xF_x$ (scale bar, 200 nm). **k** Membrane depolarization induced by Mito-$F_xF_x$ after 1 or 6 h (scale bar, 2 μm) and **l** intracellular ROS generation (scale bar, 5 μm). **m** *From left*: TEM image of Mito-GG (scale bar, 200 nm). **n** Membrane depolarization induced by Mito-VV after 1 or 6 h (scale bar, 5 μm), and **o** intracellular ROS generation. **p** TEM images of mitochondria within the HeLa cell showing the morphological damage induced by Mito-FF. Fibers formed inside the mitochondria and extruded out of the membrane (scale bar, *from left to right*; 1 μm, 200 nm, and 50 nm). **q** Mito-GG-treated mitochondria HeLa cell showing no damage of mitochondria. **r** Control HeLa cell mitochondria (scale bar, 500 nm)

Supplementary Fig. 21) and hydrophobicity of Mito-FF fibrils favors strong interactions with the mitochondrial membrane, which could destroy the negatively charged mitochondrial membrane[22, 23]. To determine whether the fibrous assembly could induce membrane disruption, we conducted a dye-leakage assay using a model liposome encapsulated with a self-quenching dye (calcein). The addition of Mito-FF fibrils (500 μM) rapidly caused a loss of membrane integrity in the liposomes, as indicated by the leakage of the encapsulated calcein dye (Fig. 5a). Dye leakage increased over time and reached a maximum leakage of 50% within 45 min. However, the addition of Mito-FF in the molecular state (i.e., below the CAC of 10 μM) did not induce liposome leakage and Mito-GG (500 μM), which formed a spherical assembly as described above, showed no liposome leakage. These results suggest that fibrils formed in the mitochondria could effectively interact with the membrane and disrupt it, leading to mitochondrial dysfunction.

To study the translocation or release of mitochondrial protein as a consequence of membrane disruption, a recently developed engineered peroxidase (APEX) labeling technique was employed[24]. For this experiment, APEX was genetically targeted to the mitochondrial matrix in live cells as confirmed by previous

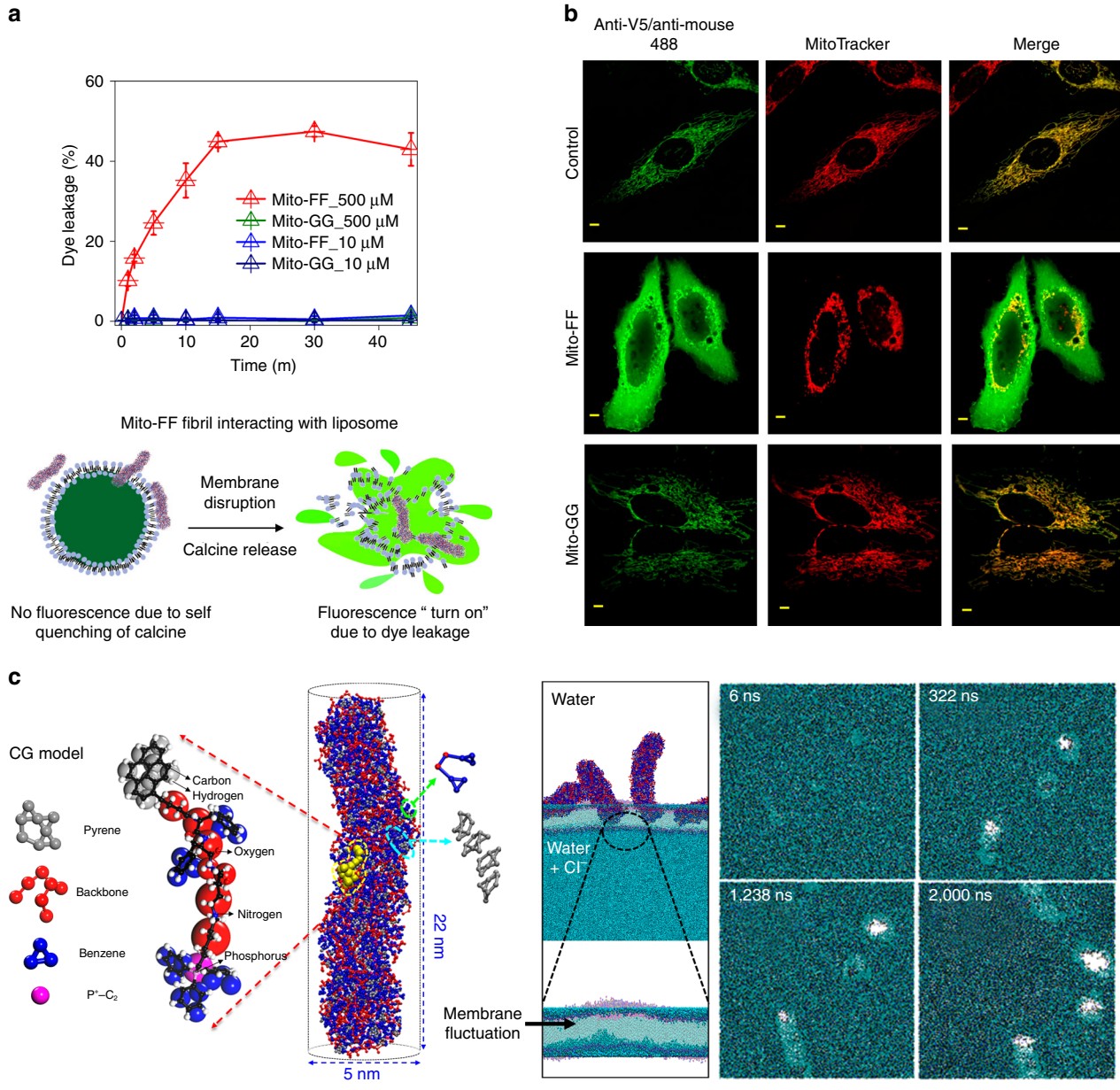

**Fig. 5** Membrane disruption induced by Mito-FF. **a** Membrane disruption ability of Mito-FF fibrils monitored using calcein encapsulated in a model liposome; a schematic diagram is provided. Data represent mean ± s.d. from three independent experiments. **b** Leakage of mitochondrial protein to the cytosol monitored using APEX labeling (scale bar, 2 μm). **c** Molecular simulation results of Mito-FF fibrils with coarse-grained (CG) model and analysis of their membrane penetration ability over time. Details of simulation method are introduced in the Supplementary Methods

electron microscope imaging experiments[25]. In our assay, we employed V5 epitope-tagged APEX2 (i.e., Mito-V5-APEX2) for immunofluorescence imaging of APEX2 with an anti-V5 antibody. We expect that the localization pattern of Mito-V5-APEX2 should be changed by Mito-FF treatment because it may compromise the integrity of the inner mitochondrial membrane (IMM) by in situ fibril formation. As expected, we could observe that Mito-V5-APEX2 was diffused from mitochondrial matrix to the cytosol after 4 h of incubation with Mito-FF (anti-V5 immunofluorescence, Fig. 5b). This result indicates that release of mitochondrial matrix proteins including Mito-V5-APEX2 occurred. We also found that severe mitochondrial fragmentization occurred upon Mito-FF treatment, which indicates that Mito-FF is considerably toxic toward mitochondrial physiology. This mitochondrial fragmentation induced by Mito-FF was also confirmed by using MitoTracker (Fig. 5b, *middle panel*).

However, incubation with Mito-GG did not cause protein release to the cytosol; indeed, the mitochondria appeared healthy even after 6 h of incubation with Mito-GG (Fig. 5b, *third panel*). This result suggests that in situ fibril formation by Mito-FF inside the mitochondria disrupted the IMM, whereas spherical assembly by Mito-GG did not compromise the integrity of the IMM. Our results do not exclude the promiscuous interact of the fibrils with mitochondrial proteins to induce mitochondrial dysfunction. However, considering that the membrane disruption occurred rapidly, i.e., within 1 h, after Mito-FF fibril formation, it is likely that promiscuous interactions of the self-assembled fiber with mitochondrial proteins do not significantly contribute to mitochondrial dysfunction.

To investigate the mechanism underlying the membrane disruption induced by Mito-FF, we investigated the self-assembly and membrane interactions of the amphiphilic peptides

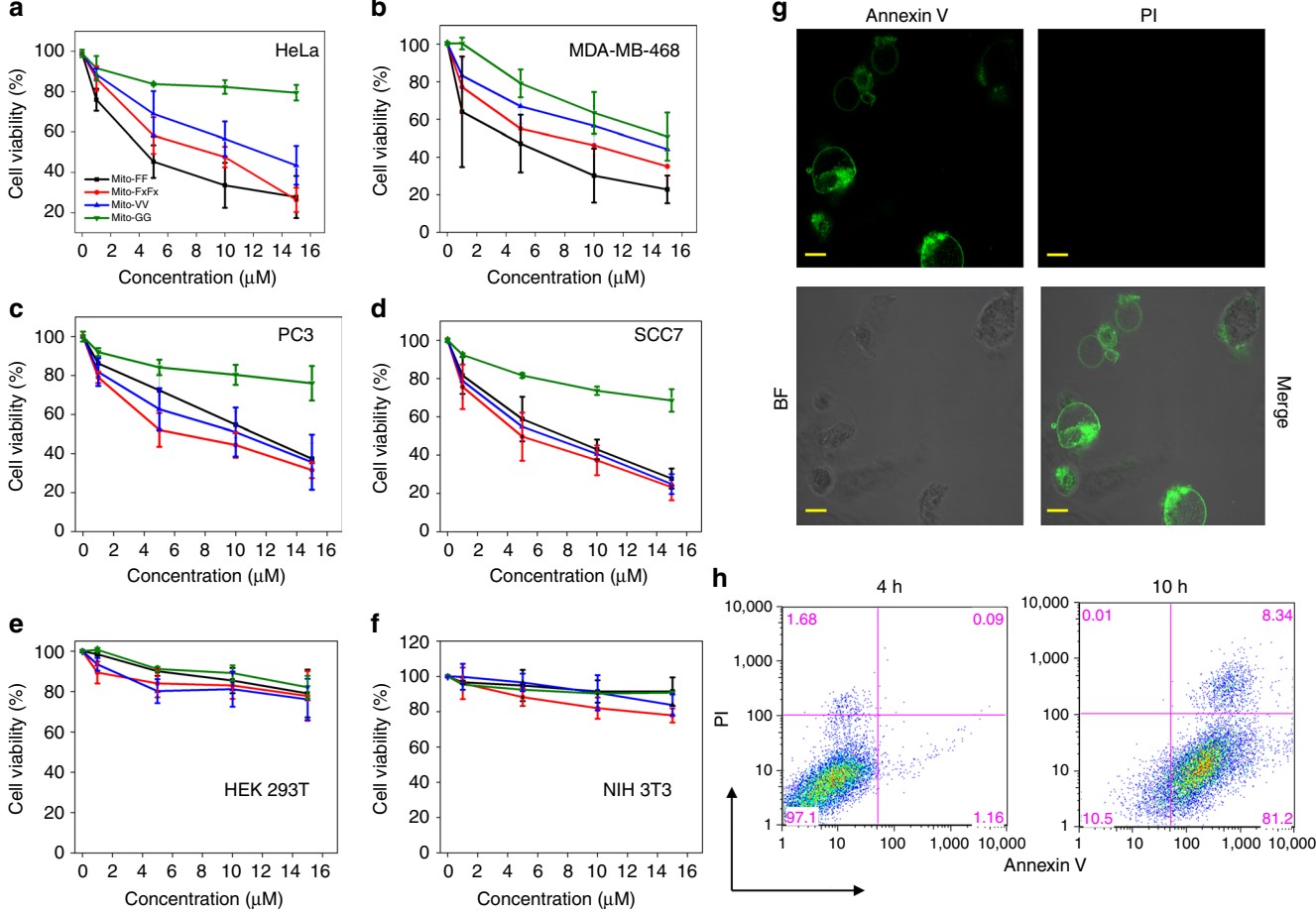

**Fig. 6** Intra-mitochondrial assembly for cancer therapy. **a–d** Analysis of Mito-FF toxicity in different cancer cell lines. Data represent mean ± s.d. from three independent experiments. **e–f** Analysis of Mito-peptide toxicity in normal cell lines. **g** Cell death induced by Mito-FF was monitored by Annexin V/PI staining, which showed early apoptosis (scale bar, 10 μm). **h** Flow cytometric analysis showing early apoptosis induced by Mito-FF

by CGMD using a Martini force field. Mito-FFs self-assembled into fibrils with an effective radius and length of ~2.5 and ~22 nm, respectively (Fig. 5c). The membrane penetrability of the Mito-FF fibrils was established using three assembled Mito-FFs, which were located above the cell membrane. A distinct hole in the cell membrane was generated by the self-assembly of the peptide, where a vertical fiber generated a hole faster than a horizontal fiber, and inflow of water occurred. To identify the main reason for the generation of the hole by the self-assembly, we traced the molecular behavior of each component of Mito-FF via root mean square fluctuation (RMSF) and the radial distribution function (RDF) (Supplementary Fig. 22). First, RMSF, which compares the current position of a particle with a reference position over time and averages the fluctuated distance for each particle, indicated that all components of Mito-FF move in the presence of the cell membrane. Phenyl groups showed the greatest fluctuation, followed by TPP ring groups. In particular, when compared with the assembly without the cell membrane, the TPP rings showed the largest increase in RMSF. Second, although RMSF determines self-interactive distances, RDF directly shows the probability of finding each component of Mito-FF around the cell membrane. We found that phenyl rings and TPP rings were highly dense around the cell membrane and play major role in the penetration, but not the backbone. In contrast, the backbone showed high-surface density with high RMSF distance and RDF peaks in Mito-GG (Supplementary Fig. 23). Based on the RMSF and RDF results, we speculated that the presence of benzene rings among the phenyl rings and TPP

rings and the high density of these structures are essential for penetration of the cell membrane.

**Intra-mitochondrial assembly for cancer therapy.** Notably, membrane-depolarization analysis in non-cancerous HEK293T cells showed no significant disappearance of TMRM intensity after incubation with Mito-FF (Supplementary Fig. 24). This indicates that Mito-FF induces mitochondrial dysfunction in HeLa, carcinoma cells, but not in normal HEK293T cells. Based on these results, we expect that our system will provide a platform for cancer-specific therapy because mitochondrial dysfunction initiates cellular death. We analyzed the toxicity of the Mito-peptides towards a variety of cancer cell lines such as HeLa, MDA-MB468 (breast), SCC 7 (skin), and PC 3 (prostate). Significant toxicity was observed for the fiber-forming peptides Mito-FF, Mito-VV, and Mito-$F_xF_x$ with a half-inhibitory concentration ($IC_{50}$) of 4–10 μM (Figs. 6a–d). However, Mito-GG had comparatively lower toxicity than its fiber-forming analog in majority of the cancer cell lines, which indicates that the mitochondrial dysfunction caused by fiber formation inside the mitochondria is responsible for the cellular death. Propidium iodide (PI) and FITC-annexin V staining assays showed annexin V staining on the plasma membrane of Mito-FF-treated HeLa cells, but PI was excluded from the cells (Fig. 6g), suggesting that the cells entered in an early apoptotic stage within 6 h of treatment with Mito-FF[26]. $H_2O_2$ (500 μM) was used as the positive necrotic control which showed red staining of PI as a result of

necrosis and Camptothecin ($20\,\mu M$) was used as a positive control for late apoptosis (stains both PI and annexin V). The membrane impermeable PI discriminates live or early apoptotic cells from late apoptotic or necrotic cells that lose membrane integrity. AnnexinV stains both apoptotic cells, which expose phosphatidylserine extracellularly, and necrotic cells, which lose membrane integrity. (Supplementary Fig. 25). Moreover, the cellular morphology with membrane blebbing after treating with Mito-FF clearly indicates that cells undergo apoptosis (Supplementary Fig. 26) Fluorescence-activated cell sorting (FACS) was conducted to quantitatively analyze the apoptosis at successive time points (2, 4, and 10 h) with the FITC-annexin V/PI staining assay (Fig. 6h and Supplementary Fig. 27). Notably, Mito-FF induced only less than 20% reduction in cell viability in normal cell lines including HEK293T and NIH 3T3, with mitochondrial features similar to those of normal cells (Fig. 6e–f). This indicates that the supramolecular approach inside the mitochondria could be used for cancer-selective therapy. Considering that the cellular membrane and mitochondrial membrane in cancer cells show much higher negative potential than those of normal cells[11], we envision that the reduced toxicity toward normal cell lines might reflect lower Mito-FF accumulation inside the mitochondria of HEK293T and NIH 3T3 cells. Indeed, the accumulation of Mito-FF inside normal cell lines was decreased to approximately 10-fold less than that in the cancer cell lines (Supplementary Fig. 30). The low Mito-FF fibril concentration inside the mitochondria might be insufficient to induce dysfunction.

Our study describes the supramolecular fibrous assembly of a peptide amphiphile inside the mitochondria of live cells. The mitochondria-specific peptide amphiphile Mito-FF accumulated inside mitochondria and formed fibrils as a result of the high local concentration, higher than the CAC, inside mitochondria. The fibrils induced drastic damage in the mitochondria by disrupting their membrane integrity and caused the mitochondrial contents including proteins to leak to the cytosol, which eventually led the cell to undergo apoptosis. As the next step of our study, we will investigate in vivo metabolic stability and pharmacokinetics of our compounds with a xenograft tumour model. Even though the membrane potential difference between the normal and cancer could bring the effective selectivity in in vitro studies, in vivo required even more specificity to cancer (i.e., conjugation of Mito-FF with cancer targeting moiety like RGD, or folic acid). If the peptides are degraded quickly before reaching cancer cells, then, it is necessary to investigate the impact of D-Phe analogue of Mito-FF to get more insight about the in vivo stability since D-amino acids are not well recognized by natural enzyme. We plan to later apply this strategy, OLISA for selective eradication of cancer and other disorders as well for the in depth investigation of cellular response and mechanism.

## Methods

**Materials**. All peptide building blocks were synthesized using an automated peptide synthesizer (CEM, liberty blue) on a $0.25\text{-}\mu M$ scale. The products were purified by high-performance liquid chromatography (HPLC, Agilent Technologies, USA) with a C 18 reverse column in ACN/Water mixture. Amino acids were purchased from Bead Tech, Korea and APEX Bio, Houston, USA. The tetra methyl rhodamine methyl ester (TMRM) was purchased from Santa Cruz Biotechnologies, Korea. MitoSox, dihydroethidium (DHE), MitoTracker Red FM, annexin V, PI, and Alexa Fluoro 488 were purchased from Life Technologies, Korea. All other chemicals used in this manuscript were purchased from Sigma Aldrich, Korea. The peptides were synthesized using automated microwave peptide synthesizer (Liberty Blue). The successful synthesis of peptides was confirmed with matrix-assisted laser desorption/ionization (MALDI-TOF/TOF, Ultraflex III). The confocal laser scanning microscopy images were taken with LSM 7800 and FV 1000 confocal microscopes. The transmission electron microscopy images were obtained using a BioTEM system (JEM 1400). The spectrofluorimetric analysis was performed using a fluorimeter (Cary Eclipse).

**Cell culture**. Human cancer cells originating from the prostate (PC3), cervix (HeLa), breast (MDA-MB-468), skin (SCC 7), as well as noncancerous fibroblast (HEK293T) and mouse embryonic fibroblasts (NIH 3T3) cells, were obtained from ATCC and cultured in DMEM, L-15, or RPMI (Life Technologies) containing 10% fetal bovine serum (FBS; Life Technologies) and 1% penicillin/streptomycin (Life Technologies) at 37 °C in a humidified atmosphere of 10% $CO_2$. For cytotoxicity analysis, cells were cultured in different concentrations of peptides (1, 2, 3, 5, 10, and $15\,\mu M$ per well). Cell viability was measured at 24 h using the Alamar blue assay, with each data point measured in triplicate. Fluorescence measurements were obtained using a plate reader (Tecan Infinite Series, Germany) by setting the excitation wavelength at 565 nm and monitoring emission from the 96-well plates at 590 nm; 96-well Nunc (Thermo Fisher Scientific Inc.) plates were seeding with cells at a density of $5 \times 10^3$ cells per well; the cells were allowed to settle for 24 h with incubation at 37 °C under 5% $CO_2$ in the respective growth medium (e.g., RPMI 1640, DMEM, or L-15).

**Imaging of ROS in mitochondria and nuclei**. HeLa and HeK293T cells were seeded on a Lab Tek II slide chamber at 80% confluence in DMEM (Life Technologies) supplemented with 10% FBS and 1% penicillin/streptomycin and incubated at 37 °C under 5% $CO_2$. After incubation with all peptides at $10\,\mu M$ for different intervals, by following the manufacturer's protocol (MitoSOX, M36008); the cell culture medium was then replaced with $1\text{--}2$ mL of $5\,\mu M$ MitoSOX reagent working solution to cover the adherent cells. The cells were then incubated for 10 min at 37 °C, protected from light. The cells were then washed gently and analyzed under a FV1000 laser confocal scanning microscope. Similarly, the dihydroethidium (DHE) assay was performed to evaluate ROS-related cytotoxicity by following the manufacturer's protocol (Santa Cruz, SC-204724A); the DHE was imaged using a FV1000 laser confocal scanning microscope.

**Imaging of apoptosis/necrosis-dependent cell death**. HeLa cells were seeded on a Lab Tek II slide chamber at 80% confluence in DMEM (Life Technologies) supplemented with 10% FBS and 1% penicillin/streptomycin at 37 °C under 5% $CO_2$. The cells were then incubated with the peptides at $10\,\mu M$ for different intervals and gently washed. The cell culture medium was then replaced with Alexa Fluor 488-conjugated annexin V and PI in 1 ml of annexin binding buffer and incubated for 15 min at 37 °C. The cells were then washed gently, the medium was replaced with DMEM medium, and the cells were analyzed using an FV1000 laser confocal scanning microscope.

**Flow cytometry analysis**. HeLa cells were seeded in a 25-mL T-flask (Thermo Scientific) in DMEM (Life Technologies) supplemented with 10% FBS and 1% penicillin/streptomycin at 37 °C under 5% $CO_2$. The cells were then incubated with the peptides at $10\,\mu M$ for a period of 1 h, 4 h, 6 h, 8 h, and 10 h. Similarly, self-assembled peptides at $100\,\mu M$ were incubated for a period of 3 h in PBS solution in triplicate. The cells were collected and washed in cold PBS. Following the manufacturer's protocol (Life Technologies, V13241), the cells were incubated with 5 µl of Alexa Fluor 488-conjugated annexin V and 1 µl of $100\,\mu g\,ml^{-1}$ PI working solution in 100 µl of annexin-binding buffer solution for 15 min at room temperature. After the incubation period, 400 µl of 1× annexin-binding buffer was added with gentle mixing, and the samples were kept on ice. The stained cells were then analyzed by flow cytometry (FACS ealibur, BD Bioscience), with the fluorescence emission measured at 530 nm (i.e., FL1) and >575 nm (i.e., FL3).

**Data availability**. The data that support the findings of this study are available from the authors on reasonable request, see author contributions for specific data sets.

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

## Acknowledgements

This work was supported by the Korea Foundation for the Advancement of Science & Creativity (KOFAC), and funded by the National Research Foundation of Korea (MSIP 2014R1A1A1002642) and by the Basic Science Research Program (2016R1A2B4012322, 2014R1A5A1009799, and 2016R1A5A1009405) through the National Research Foundation of Korea (NRF), and by the Ulsan National Institute of Science and Technology research fund (Grant 1.160001.01). Computational resources are from UNIST-HPC and KISTI-PLSI. We thank Prof. B.H. Kang at UNIST for kindly providing isolated mitochondria.

## Author Contributions

J.-H.R. initiated the work. M.T.J. and J.-H.R. conceived and designed the experiments. M.T.J. synthesized and characterized the compounds, studied the self-assembly behaviors, performed confocal experiments, dye leakage assay, mitochondrial accumulation and cell viability analysis. L.P. performed flow cytometric analysis, mitochondria isolation and cell viability tests. E.M.G. and S.K.K. performed molecular simulations. M.-G.K. and H.-W.R. performed protein leakage experiments. I.K., S.-M.J and E.L. performed the TEM experiments, S.L. and S.C.B. performed two photon microscopy imaging. C.K. prepared the cell TEM samples, S.P. and H.C. contributed to it. M.T.J. and J.-H.R. wrote the manuscipt with the contribiutions from E.M.G., S.K.K. (Simulation part); I.K., S.J.M., E.L. (TEM part); M.-G.K., H.-W.R. (Protein analysis part).

## Additional information

**Competing interests:** The authors declare no competing financial interests.

