## [Peer Review File · Nature Communications]

Reviewers' comments:

Reviewer #1 (Remarks to the Author):

This paper reports on localized self-assembly of peptide derivatives in mitochondria, which allows for a new way to control cellular fate. What is conceptually novel about this work is that it is focused on raising local concentrations of the self-assembling molecules by delivering them to an organelle (in this case mitochondria) without having to rely on additional stimuli- the entire process is controlled by intracellular trafficking and localized accumulation. Once localized, the critical assembly concentration is eventually reached and upon formation of fibrils (but not spheres) mitochondria are disrupted and selective cell death results. Interestingly, the treatment impacts cancer cells more than normal cells, which is proposed to be related to the differential membrane potential of mitochondria in cancer cells. Overall, this paper provides an important new concept in cancer therapeutics and it is convincingly demonstrated.

Very recently, a paper with some conceptual overlap was published by Bing Xu et al. in JACS. I would like to stress that the approach followed in that paper however relies on the enzymatic assembly at the cell surface where the supramolecular assembly takes place, which is followed by uptake and targeting of mitochondria. The organelle localization approach presented in this paper is more direct and general in that it does not rely on enzymatic assembly and does not require assembled structures to be transported in the cell- they are formed at the target site. As a result much lower dosage of molecules is required which is a key issue with the enzymatic activation method.

The mode of action is demonstrated convincingly using confocal microscopy and TEM analysis as well as consideration of the concentration accumulation upon localization in the mitochondria and critical assembly concentration and supramolecular shape of structures formed. The paper is exciting and important, it is clearly written and can be accepted provided that the following minor issues are addressed.

Major:

For the paper to reach the impact that it deserves it is important that the difference with the EISA approach is clearly explained, with reference to the above mentioned Xu paper.

Minor:

1. The authors state "However, achieving spatiotemporal control (i.e., inside cellular organelles or other sub-compartments) over the self-assembly of synthetic molecules inside the cell is challenging because of the difficulty of studying their behavior in the complex intracellular environment, and thus has not yet been reported." This is not strictly true as Bing Xu has reported accumulation in pericellular space.
2. Refs 4 and 6: Uljin should be Ulijn

Reviewer #2 (Remarks to the Author):

This manuscript tells an interesting story and is elegant in its experimental approach – I enjoyed reading the paper. The experimental data are extensive and are backed up by a range of supplementary information. The outcomes are novel, interesting and of significance to the field.

Some comments:

On page 3, the fibres are described as being 9-10 nm in length. How many fibres and how many TEM images were measured to come to this conclusion? A description of the methodology applied

(e.g. n =100 fibrils, X image, measurement tool etc) would be helpful. The image in Figure 2 b suggests the fibrils could be a bit thicker than 10 nm.

The peptides Mito-FF and Mito-FF-NBD are described as co-aggregating and forming the same fibril. MD simulations and TEM images are used as evidence that such co-aggregation occurs (Supplementary Figures 9 and 10). The legend of the MD simulations requires some further explanation. It is also not clear why the TEM image indicates co-aggregation. Do the fibrils differ in width to those formed by Mito-FF or Mito-FF-NBD alone? No interpretation is given as evidence to support the conclusions drawn in the main text.

Mito-FF is thought to assemble within the mitochondria as it reaches a concentration greater than the CAC. It is proposed that Mito-FF-NBD doesn't assemble as its concentration does not reach the CAC. Why would this be the case? What limits the concentration of the second molecule within the mitochondria? Can the authors suggest a possible reason for the difference?

I struggle to see the fibrils of 8 nm width within the TEM images of the mitochondria from the mouse brain in Figure 3f-g. The arrows aren't particularly helpful in distinguishing the internal membrane structures of the mitochondria from the fibrils. I think that the fibrils are the diffuse grey stained areas. Do the authors have a higher resolution image? Could a digital zoom be useful here? Or could insets and better labelling of arrows be used to distinguish between the structures. The authors claim that the fibrils in these images are 8 nm in width but how have they made these measurements to give this accuracy? Is this by visual inspection or by a series of measurements using a tool and if so what was the error of the measurement?

Confocal microscopy is used for many of the images presented in the paper but curiously only 2D images are presented. Did the authors collect any 3D images of the fibrils within the cells? This would provide useful additional information to assist readers to understand the 3D localisation of the fibrils within the mitochondria and cells and would be a helpful addition.

The Mito-FF fibrils are suggested to penetrate the membrane, as they appear on the outside of the cell membrane in image 4p of mouse cells. It is very difficult to see any evidence of these fibrils within these cells or across the membrane and an alternative hypothesis could be that in this instance the fibrils had assembled on the outside of the cell. I suggest more moderate language be used to describe the possibility of the fibrils penetrating as this is not really direct evidence. A time series or image that showed fibrils either side might be more conclusive.

The high positive charge and hydrophobicity of the Mito-FF surface are described as being key reasons behind the proposed mechanism. Have these material properties been described previously? If so can a reference be provided? Alternatively can the characterisation data be presented?

Propidium iodide is shown to be excluded from the cells treated by Mito-FF in Figure 6g. As there is very little staining in this image it would be good to include a control image that shows positive PI staining using say an apoptotic chemical and a later stage of apoptosis.

The references to supplementary Figure 18, 19 and 20 appear to be missing from the main text. Without these references or discussion there doesn't seem much point in including these Figures.

Supplementary Figure 22 indicates that the peptides accumulate in cancerous cells but to a lower extent in normal cells. Why would this be the case? Most of this work has also been produced on cell lines. Were primary isolates examined?

The work is promising but have the authors considered the next steps in the evaluation of their OLISA approach? Once the cell undergoes apoptosis what are the likely clearance rates for these molecules and persistence in the body? Is anything known for these molecules or for molecules

like those of this study? A comment would be worthwhile.

Minor points:

In Figure 2 c there are a number of bands assigned and highlighted within the spectra but only one is discussed in the text. Are the other bands significant? If so can they be described?

On page 4 – the description of ‘similar uptake’ at 4°C is confusing. Relative to what? Perhaps this just needs rephrasing.

On page 4 Mito-FF is described as diffusing through the plasma membrane. Is there any prior information on the diffusion of FF structures through the membrane? A reference would be useful here.

On page 6 – significant figures appear to be overstated e.g. 10.331 mM internal cell Mito-FF concentration.

It is worthy of comment in the main text that the intensity of staining in supplementary Figure 11 is quite low – i.e. there is only limited evidence of ROS production via this method.

In Figure 6 and Supplementary Figure 21- the labelling of the dye on each axes for the FACS appears missing from the legend.

Reviewers' comments:

Reviewer 1 (Remarks to the Author):

This paper reports on localized self-assembly of peptide derivatives in mitochondria, which allows for a new way to control cellular fate. What is conceptually novel about this work is that it is focused on raising local concentrations of the self-assembling molecules by delivering them to an organelle (in this case mitochondria) without having to rely on additional stimuli- the entire process is controlled by intracellular trafficking and localized accumulation. Once localized, the critical assembly concentration is eventually reached and upon formation of fibrils (but not spheres) mitochondria are disrupted and selective cell death results. Interestingly, the treatment impacts cancer cells more than normal cells, which is proposed to be related to the differential membrane potential of mitochondria in cancer cells. Overall, this paper provides an important new concept in cancer therapeutics and it is convincingly demonstrated.

Very recently, a paper with some conceptual overlap was published by Bing Xu et al. in JACS. I would like to stress that the approach followed in that paper however relies on the enzymatic assembly at the cell surface where the supramolecular assembly takes place, which is followed by uptake and targeting of mitochondria. The organelle localization approach presented in this paper is more direct and general in that it does not rely on enzymatic assembly and does not require assembled structures to be transported in the cell- they are formed at the target site. As a result much lower dosage of molecules is required which is a key issue with the enzymatic activation method.

The mode of action is demonstrated convincingly using confocal microscopy and TEM analysis as well as consideration of the concentration accumulation upon localization in the mitochondria and critical assembly concentration and supramolecular shape of structures formed. The paper is exciting and important, it is clearly written and can be accepted provided that the following minor issues are addressed.

Major:

Comment 1:

For the paper to reach the impact that it deserves it is important that the difference with the EISA approach is clearly explained, with reference to the above mentioned Xu paper.

Author Response 1.

Thank you for the valuable comment. We agree with the comment. It is important to provide a clear explanation about how organelle localization induced self-assembly (OLISA) is different from other intracellular assembly approaches such as enzyme instructed intracellular assembly (EISA). Briefly, in EISA, an enzyme changes the molecular structure of the precursor from soluble hydrophilic to self-assembling units to form ordered-structure within the cell or pericellular space which is a challenging approach and induces cancer cell death effectively. However, EISA could not be generalized to all cancer cells, since it always required an enzyme to induce assembly which has different expression in different cell lines. Moreover, the EISA occurs via an enzymatic conversion (chemical reaction involving bond-

breaking) inside the cell or near the cell surface, which is a time-consuming process, and thus the time required for its apoptotic impact to the cell is much higher (~ 48 h) (Xu *et al. J. Am. Chem. Soc.* **138**, 16046-16055 (2016)). OLISA is more general and simple strategy without any complicated chemical reactions. In here, the small molecules readily diffuse through the cell membrane, reach to the target site (organelle or sub-cellular compartment depending on the targeting moiety) and then they undergo self-assembly inside the targeted organelle as a result of increased local concentration. The accumulation of molecules inside an organelle like mitochondria is ~ 500 – 1000 times higher than that of extracellular space (Chandel *et al. Nat. Chem. Biol.* **11**, 9-15 (2015)). While EISA which is usually happened inside the cell or near the cell surface requires very high concentration of molecules (over several hundreds micromole), OLISA occurs with low dosage concentration (several tens micromole), which is a superior advantage of OLISA. We describe the difference between EISA and OLISA in revised manuscript like following:

Revised version in the manuscript, page 2– page 3:

Recently, enzyme instructed intracellular self-assembly (EISA) has emerged as an effective approach to induce cellular dysfunction, in which an enzyme changes the molecular structure of precursor from soluble hydrophilic to self-assembling units to form ordered-structure within the cell or pericellular space^{8,9}. However, EISA could not be generalized to various cancer cells, since the usage of biological stimuli (e.g., enzymes) is usually limited to specific cell types or cellular compartments. Moreover, the EISA occurs via an enzymatic conversion (chemical reaction involving bond-breaking) inside the cell or near the cell surface, which is a time-consuming process.

We here hypothesize that a specific cellular organelle-localization induced supramolecular self-assembly (OLISA) system could be a general strategy to induce self-assembly by raising local concentrations of the self-assembling molecules without additional treatment. The small molecules readily diffuse through the cell membrane, reach to the target site (organelle or subcellular compartment depending on the targeting moiety) and then they undergo self-assembly inside the targeted organelle as a result of increased local concentration. The accumulation of molecules inside an organelle like mitochondria is ~ 500 – 1000 times higher than that of extracellular space¹⁰. While EISA which is usually happened inside the cell or near the cell surface requires very high concentration of molecules (over several hundreds micromole), OLISA occurs with low dosage concentration (several tens micromole), which is a superior advantage of OLISA.

Minor:

Comment 2:

1. The authors state “However, achieving spatiotemporal control (i.e., inside cellular organelles or other sub-compartments) over the self-assembly of synthetic molecules inside the cell is challenging because of the difficulty of studying their behavior in the complex intracellular environment, and thus has not yet been reported.” This is not strictly true as Bing Xu has reported accumulation in pericellular space.

Author Response 2.

We agree with the comment. As per the reviewer’s suggestion we have changed the sentence

in the manuscript as follows:

Revised version in the manuscript, page 2:

However, achieving spatiotemporal control (i.e., inside cellular organelles or other sub-compartments) over the self-assembly of synthetic molecules inside the cell is challenging because of the difficulty of studying their behavior in the complex intracellular environment.

Comment 3:

2. Refs 4 and 6: Ulijn should be Ulijn

Author Response 3.

We apologize for the typo error. We have changed the reference 2, 4 and 6 as follows in the manuscript.

Reference 2

2. Ulijn, R. V. Smith, A. M. Designing peptide based nanomaterials. *Chem. Soc. Rev.* **37**, 664-675 (2008).

Reference 4.

4. Frederix, P. W. J. M., Scott, G. G., Abul-Haija, Y. M., Kalafatovic, D., Pappas, C. G., Javid, N., Hunt, N. T., Ulijn, R. V & Tuttle, T. Exploring the sequence space for (tri-) peptide self-assembly to design and discover new hydrogels. *Nat. chem.* **7**, 30-37 (2015).

Reference 6.

6. Pires, R. A., Abul-Haija, Y. M., Costa, D. S., Novoa-Carballal, R., Reis, R. L., Ulijn, R.V & Pashkuleva, I. Controlling cancer cell fate using localized biocatalytic self-assembly of an aromatic carbohydrate amphiphile. *J. Am. Chem. Soc.* **137**, 576-579 (2015).

Reviewer 2 (Remarks to the Author):

This manuscript tells an interesting story and is elegant in its experimental approach – I enjoyed reading the paper. The experimental data are extensive and are backed up by a range of supplementary information. The outcomes are novel, interesting and of significance to the field.

Some comments:

Comment 1.

On page 3, the fibers are described as being 9-10 nm in length. How many fibers and how many TEM images were measured to come to this conclusion? A description of the methodology applied (e.g. n =100 fibrils, X image, measurement tool etc.) would be helpful. The image in Figure 2 b suggests the fibrils could be a bit thicker than 10 nm.

Author Response 1.

We thank for the valuable comments. As described in the manuscript, **Mito-FF** self-assembles into nanofibers with a width of 9–10 nm for the individual fibers and a length of several micrometers in phosphate-buffered saline (PBS). Reviewer points out that the image in Fig. 2b suggests the fibrils could be a bit thicker than 10 nm. We apologize for the scale bar error in the Fig. 2b, that was a mistake happened during the conversion. The scale bar corrected image is given below, which is included as the Supplementary Fig. 3. We have measured the diameter of 100 nanofibers with more than 5 TEM micrographs using simple measure program (JEOL Ltd., Tokyo, Japan). The applied methodology is included in the supplementary information (Supplementary methods, 1.3) as given below. Additionally, we have replaced Fig. 2b in the manuscript to show clear.

We have modified the manuscript with below information.

Revised manuscript, page 4.

Mito-FF self-assembled into nanofibers with a width of 9.6 ± 1.1 nm averaged over 100 nanofibers for the individual fibers and a length of several micrometers in phosphate-buffered saline (PBS).

Revised supplementary methods, page 2.

1.3 Transmission Electron Microscopic analysis of **Mito-FF**.

A drop of the aqueous solution of **Mito-FF** was placed on a formvar/carbon-coated copper grid and allowed to evaporate under ambient conditions. The sample was stained with 2 wt% uranyl acetate solution, allowed to evaporate for 1 min and excess solution removed with a filter paper. The specimen was observed with JEM-1400 TEM operating at 120 kV. Structural information of self-assembled **Mito-FF** fibrils was obtained with over 100 nanofibers of more than 5 micrographs by simple measure program (JEOL Ltd., Tokyo, Japan).

Revised Figure 2: Figure 2b is replaced.

Figure 2 | Self-assembly and mitochondrial localization of Mito-FF. (b) TEM image of Mito-FF fibril in water (scale bar: 50 nm).

Revised Supplementary Figure 3:

Supplementary Figure 3 | a) TEM images (stained with 2 wt% uranyl acetate) for Mito-FF fibers and b) diameter distribution.

Comment 2.

The peptides *Mito-FF* and *Mito-FF-NBD* are described as co-aggregating and forming the same fibril. MD simulations and TEM images are used as evidence that such co-aggregation occurs (Supplementary Figures 9 and 10). The legend of the MD simulations requires some further explanation. It is also not clear why the TEM image indicates co-aggregation. Do the fibrils differ in width to those formed by *Mito-FF* or *Mito-FF-NBD* alone? No interpretation is given as evidence to support the conclusions drawn in the main text.

Author Response 2: [MD simulation part]

We have included more explanation for the molecular simulations in the Supplementary Information. In order to provide clear explanation of the model system and CGMD simulation for **Mito-FF-NBD** co-aggregation, we have modified Supplementary Fig. 10 and revised captions. Simulation details are given in the section 1.6. in supplementary methods. We have also modified Fig. 5c in the original manuscript to provide better resolution.

Revised Supplementary Figure 10.

Supplementary Figure 10 | Coarse-grained (CG) model of (a) **Mito-FF-NBD** and (b) **Mito-FF**. Details of simulation method are introduced in the section 1.6. NBD, phenyl ring, TPP and pyrene groups consist of six, six, ten and eight CG beads, respectively. NBD group contains O–N–O, N–O–N, N–C₃–O and benzene ring. Backbone of **Mito-FF-NBD** consists of five CG beads, which are 3 amides and 2 amines, and **Mito-FF** consists of seven CG beads, which are 1 ketone, 3 amides, 2 amines, and 1 alkyl groups. (c) Snapshot of the simulation result of cylindrically self-assembled 174 **Mito-FF-NBDs** and 174 **Mito-FFs** in the box (i.e. 25×25×30 nm³) filled with water after performing CGMD for 3.6 μs. The cylinder shows effective radius and length of the **Mito-FF-NBD** and **Mito-FF** fibril. (d) Radial number density of four constituent molecules in the fibril from the principal axis of fibril to its surface.

The numbers of groups in the same volume of radial shell were counted six times within the final 50 ns of the MD simulation and averaged with a 10 ns interval. Hydrophilic TPP and hydrophobic phenyl groups showed high density compared to others since they were contained in both **Mito-FF-NBD** and **Mito-FF**. High density of phenyl group at the center of fibril indicated the hydrophobic characteristics inside of the fibril. NBD groups were well distributed in that hydrophobic environment within the fibril.

Revised supplementary methods, page 4.

1.6 Molecular Dynamics Methods:

In order to perform coarse-grained molecular dynamics (CGMD) simulation, a CG force field is required. Martini force field was developed by Marrink *et al.* for lipid membrane, and recently, the coverage of the force field has been extended to proteins and carbon-based molecules^{1,2}. The coarse-graining basic rule in Martini force field is that two to four carbon-compositions are coarse-grained to one bead. For example, C₄H₁₀ can be represented as one bead and benzene ring can be represented as three beads, where each bead contains C₂H₂ group. To incorporate Martini force field to our system, **Mito-FF-NBD**, **Mito-FF** and **Mito-GG** were coarse-grained by following the rule of Martini as shown in Supplementary Figs. 9a, 9b and 16a, respectively. For the self-assembly simulations of **Mito-FFs** and **Mito-GGs**, 243 molecules were used in the box of 25×25×30 nm³ and 30×30×30 nm³, respectively, which were filled with water. The simulations were run for 5 μs.

For the cell membrane system with **Mito-FF** as shown in Fig. 5c, the bilayer was constructed with 3500 DPPCs and 1500 DPPGs (i.e. 2.3:1)³. We put two bilayers to separate inside and outside of the cell, where 5657 Cl⁻ ions were introduced between the bilayers to incorporate a low pH environment (i.e. ~3)⁴. The distance between two bilayers was set to 30 nm and three **Mito-FFs** were introduced at 1 nm above the membrane. For the cell membrane system with **Mito-GG** as shown in Supplementary Fig. 16c, the bilayer was constructed with 5380 DPPCs and 2306 DPPGs (i.e. 2.3:1)³. In this system, 8450 Cl⁻ ions were used to incorporate a low pH environment (i.e. ~3)⁴. The distance between two bilayers was set to 35 nm and three **Mito-GGs** were introduced at 1 nm above the membrane. The box sizes of the cell membrane systems with **Mito-FFs** and **Mito-GGs** were 30×30×72 nm³ and 50×50×80 nm³, respectively. The simulations were run for 2 μs and 1.56 μs for **Mito-FF** and **Mito-GG** systems, respectively. Note that CG models for DPPC and DPPG in the membrane system were taken from Tian and Ma's work³.

All CGMD simulations were performed at 300 K and 1 bar with Berendsen thermostat and barostat. For the cell membrane systems, semi-isotropic pressure coupling was used in the perpendicular direction to the cell membrane. The cut off radius of van der Waals and short-ranged Coulombic interaction were set to 1.2 nm. The time step was set to 20 fs. For running CGMD simulations, GROMACS 5.0.3 package was used⁴.

Revised Figure 5: Fig. 5c is replaced with new one for better clarity.

Figure 5 | (c) Molecular simulation of **Mito-FF** fibrils with coarse-grained (CG) model and analysis of their membrane penetration ability over time. Details of simulation method are introduced in the section 1.6 in SI.

Revised version in the manuscript, page 14-15

We found that phenyl rings and TPP rings were highly dense around the cell membrane and play major role in the penetration, but not the backbone. In contrast, the backbone showed high surface density with high RMSF distance and RDF peaks in **Mito-GG** (Supplementary Fig. 23).

Author Response 2: [TEM measurement part]

We apologize for the lack of enough description in the manuscript about the co-assembly of **Mito-FF** with **Mito-FF-NBD**. **Mito-FF-NBD** co-assembles with **Mito-FF** to form fiber

whose diameter is smaller than **Mito-FF** alone. To make clear, we compared the diameters of each sample measured by TEM (Supplementary Fig. 11) (more than 100 nanofibrils with over 5 TEM images for each one). From the TEM images, we observe that the diameter of co-assembled nanofibrils (7.1 ± 0.8 nm) is narrower compared with only **Mito-FF** (9.6 ± 1.1 nm). We have replaced the co-aggregation TEM image in the Supplementary Fig. 11 with new one as follows. Moreover, **Mito-FF-NBD** at the same concentration that is used for co-assembly with **Mito-FF** (10 μ M, below its CAC of 1.5 mM) does not exhibit any nanostructure.

Modified Supplementary Figure 11.

Supplementary Figure 11 | Negatively stained TEM image of a) **Mito-FF** and b) co-assembly of **Mito-FF** (500 μ M) and **Mito-FF-NBD** (100 μ M) (inset) Diameter distribution graph of nanofibrils. The image was taken with 2 wt% uranyl acetate staining. The TEM images of **Mito-FF** and **Mito-FF-NBD** co-assembly showed a decreased diameter (7.1 ± 0.8 nm averaged over 100 nanofibers) compared to **Mito-FF** nanofibrils alone suggesting that they assemble each other.

Revised version in the manuscript, page 7.

The co-assembly of **Mito-FF-NBD** with **Mito-FF** into fiber assembly in PBS was confirmed by molecular simulation (CGMD) (Supplementary Fig. 10) and TEM analysis (Supplementary Fig. 10). The TEM showed that the co-assembled fibers have a decreased diameter (7.1 ± 0.8 nm averaged over 100 nanofibers) compared to **Mito-FF** nanofibrils alone, suggesting that they assemble each other.

Comment 3.

Mito-FF is thought to assemble within the mitochondria as it reaches a concentration greater than the CAC. It is proposed that Mito-FF-NBD doesn't assemble as its concentration does not reach the CAC. Why would this be the case? What limits the concentration of the second molecule within the mitochondria? Can the authors suggest a possible reason for the difference?

Author Response 3

We thank for the valuable comment. We apologize that there was lack of explanation of co-assembly in the original manuscript. Throughout our experiments, we found that **Mito-FF** was readily uptaken by the cancer cells (within 30 min) mitochondria and the local concentration was increased over 500-1000 times with respect to the dosage concentration (10 μ M). However, we observed that **Mito-FF-NBD** under same incubation concentration as that of **Mito-FF** did not induce intra-mitochondrial self-assembly of the molecules as indicated by the lack of green fluorescence inside the mitochondria. In order to study the self-assembly behavior of **Mito-FF-NBD** inside mitochondria, we have performed a dosage dependent confocal microscopy analysis for **Mito-FF-NBD**, where we have varied the incubation concentration of **Mito-FF-NBD** as 10, 20, 40 and 100 μ M (Supplementary Fig. 12). No fluorescence observed for 10 μ M, negligible fluorescence for 20 μ M, and bright green fluorescence starts appearing above 40 μ M incubation concentration which overlap well with red fluorescence of MitoTracker. This result suggests that much higher concentration is needed for **Mito-FF-NBD** to raise intra-mitochondrial concentration above the CAC compared with **Mito-FF**. The CAC of **Mito-FF-NBD** (Supplementary Fig. 13) was found to be about 1.5 mM which is almost 25 times higher than that of **Mito-FF** (60 μ M). To get more insight about this observation, we have calculated the intra-mitochondrial concentration (IMC) of **Mito-FF-NBD** using the same method as described in the manuscript. Our result showed very low cellular uptake for 10 μ M, and the high dosage concentration (100 μ M) showed higher cellular uptake to be the mitochondria accumulation of 2.4 mM which is above their CAC (Supplementary Fig. 14). Overall, these results indicate that **Mito-FF-NBD** could co-assemble well with **Mito-FF**, but **Mito-FF-NBD** without **Mito-FF** could not self-assemble into the fiber because it could not reach to the high CAC of 1.5 mM in the mitochondria with an external concentration of 10 μ M.

Revised manuscript version, page 8.

Mito-FF-NBD could form fibrils above their CAC as shown in Supplementary Fig. 12. We observed that **Mito-FF-NBD** showed a concentration dependent fluorescence inside the cell. No fluorescence observed for 10 μ M dosage concentration, negligible fluorescence for 20 μ M and bright green fluorescence starts appearing above 40 μ M, which well-overlaps with red fluorescence of MitoTracker as indicated by the 2D and 3D confocal images (Supplementary Fig. 13 and 14). It indicates that much higher concentration is needed for **Mito-FF-NBD** to rise intra-mitochondrial concentration above the CAC compared with **Mito-FF**. When we have analyzed the IMC of **Mito-FF-NBD** in the HeLa cells by using the same procedure as described for **Mito-FF**, we found very low cellular uptake for 10 μ M while the IMC increased to 2.3 mM at high dosage concentration (100 μ M). These results suggest that **Mito-FF-NBD** could co-assemble well with **Mito-FF** and NBD located inside the hydrophobic fiber to emit bright green fluorescence. However, **Mito-FF-NBD** without **Mito-FF** could not self-assemble into the fiber because it could not reach to the high CAC of 1.5 mM in the mitochondria with an external concentration of 10 μ M.

Revised Supplementary Figure 12.

Supplementary Figure 12 | CAC determination for **Mito-FF-NBD** using Nile red encapsulation method. (a) Emission spectra of Nile red at an excitation of 550 nm. (b) Intensity of Nile red emission at 653 nm plotted against log concentration of **Mito-FF-NBD**. (c) TEM images showing the nanofibrils formed by **Mito-FF-NBD** at 2 mM concentration.

Revised Supplementary Figure 13.

Supplementary Figure 13 | Dosage dependent confocal microscopic analysis for **Mito-FF-NBD**. Blue: Nuclei stain, Red: MitoTracker, Green: **Mito-FF-NBD**.

Supplementary Figure 14 | a) 2D image showing co-localization of **Mito-FF-NBD** inside mitochondria and b) 3D localization for **Mito-FF-NBD** (middle: zoom view, right: orthogonal view). Inset: co-localization between mitotracker and **Mito-FF-NBD**. Blue: Nuclei stain, Red: MitoTracker Red FM, Green: **Mito-FF-NBD** (left)Zoom image (middle) Orthogonal view (right).

Revised Supplementary Figure 15.

Supplementary Figure 15 | IMC determination for Mito-FF-NBD. (a) Emission spectra of **Mito-FF-NBD** in Buffer/MeOH mixture (1:1) recorded for the generation of calibration plot. (b) Calibration plot of **Mito-FF-NBD** from the emission spectra. (c) Emission spectra of HeLa cell lysate (Buffer/MeOH mixture (1:1)) after 3 h treatment with **Mito-FF-NBD**. (d) IMC of **Mito-FF-NBD** after 3 h treatment with **HeLa** cells.

Comment 4.

I struggle to see the fibrils of 8 nm width within the TEM images of the mitochondria from the mouse brain in Figure 3f-g. The arrows aren't particularly helpful in distinguishing the internal membrane structures of the mitochondria from the fibrils. I think that the fibrils are the diffuse grey stained areas. Do the authors have a higher resolution image? Could a digital zoom be useful here? Or could insets and better labelling of arrows be used to distinguish between the structures. The authors claim that the fibrils in these images are 8 nm in width but how have they made these measurements to give this accuracy? Is this by visual inspection or by a series of measurements using a tool and if so what was the error of the measurement?

Author Response 4.

We agree with the concern that the **Mito-FF** fibrils in the mitochondria are hard to differentiate from the mitochondria internal membrane or crista. To distinguish fibrils from

the mitochondrial membrane, we have conducted TEM tomography (TEM T), which is a powerful method to study the internal architecture of the cellular structure or macromolecular complexes (Newmeyer *et al. J. Cell Biol.* **150**, 1027-1036 (2000)). Depending upon the three dimensional (3D) orientation at different tilt angles, we could easily differentiate the fibrils which are cylindrical in shape from the membranes which have a lamellar structure. Using TEM T we figure out the fibrils within the mitochondria with respect to tilt angles from -68° to $+68^\circ$, where at -60° focused fibril has observed in its surface view and at $+20^\circ$ the same fibril has observed from the top as shown in the Fig. R1 (below). We have included this information in the manuscript, and we changed the Fig. 3g in the original manuscript with the TEM T images. Additionally, we have replaced the Fig. 3f (control mitochondria) in the revised manuscript for clarity. To make clear that the fibrils are observed with 9.0 ± 1.5 nm in the isolated mitochondria after treating with **Mito-FF**, we have counted 100 nanofibers within the mitochondria of over 10 micrographs by Simple Measure program (JEOL Ltd., Tokyo, Japan). We have included the description in the revised manuscript and also included supplementary methods as 1.9 in Supplementary Information, also the diameter information is provided as the Supplementary Fig. 16 along with TEM images.

Figure R1 | a) 3D volume rendering of mitochondria with **Mito-FF** fibrils and b) TEM images of mitochondria at different tilt-angles of -60° , -40° and $+20^\circ$. The dotted square indicates the **Mito-FF** fibrils.

Revised version in the manuscript page 8.

The TEM images showed fibrillar structures with a diameter of 9.0 ± 1.5 nm averaged over 100 nanofibers within the mitochondria (Supplementary Fig. 16). For clear demonstration of the nanofibers within the mitochondria, TEM tomography (TEM T) was performed. TEM T is a powerful method to study the internal architecture of the cellular structure or macromolecular complexes¹⁷. In here, depending upon the three dimensional (3D) orientation of the fibrils within the mitochondria and internal membrane at different tilt angles, we could differentiate the fibrils which are cylindrical in shape from the internal mitochondrial membranes which have a lamellar structure. Using TEM T we figure out the fibrils within the mitochondria with respect to tilt angles from -68° to $+68^\circ$, where at -60° focused fibril has observed in its surface view and at $+20^\circ$ the same fibril has observed from the top as shown in the Fig. 3h (-60° (left), $+20^\circ$ (right)).

Revised supplementary methods, page 5.

1.9 TEM analysis of **Mito-FF** fibrils within the isolated mitochondria.

The specimen of **Mito-FF** fibrils within the isolated mitochondria was observed with a JEM-1400 operating at 120 kV. Structural information of self-assembled **Mito-FF** was provided with more than 100 nanofibers of over 10 micrographs by Simple Measure program (JEOL Ltd., Tokyo, Japan).

1.10 TEM Tomography (TEM T) of mitochondria with **Mito-FF** fibrils.

3D reconstruction of mitochondria obtained from a series of 2D image projections of the samples at different viewing angles. For TEM T, images were obtained using a JEM-1400 (JEOL Ltd., Tokyo, Japan), operating at 120 kV, and a charge-coupled device (CCD) camera size of 1046×1046 (JEOL Ltd., Tokyo, Japan). In total, 137 images were acquired at tilting angles between -68° and $+68^\circ$, with an increment of 1° . The magnification was $\times 30K$, corresponding to a pixel size of 0.85 nm. Tilting, refocusing, and repositioning were carried out after every individual tilt increase. Alignment and reconstruction of tilt series were performed in IMOD software. UCSF chimera software was used for visualization.

Revised Supplementary Figure 16.

Supplementary Figure 16 | a) **Mito-FF** fibrils inside mitochondria isolated from a C3H female nude mouse brain and b) diameter distribution of **Mito-FF** nanofibrils within the isolated mitochondria.

Revised manuscript figure 3: figure 3 f-g is replaced.

Figure 3 | Intra-mitochondrial assembly. (f) TEM images for isolated mitochondria (control). (g) 3D volume image of **Mito-FF** fibrils inside mitochondria isolated from a C3H female nude mouse brain, and construction images showing the side and cross-sectional views of fibrils at different tilt-angles of -60° and $+20^\circ$ (from left to right).

Comment 5.

Confocal microscopy is used for many of the images presented in the paper but curiously only 2D images are presented. Did the authors collect any 3D images of the fibrils within the cells? This would provide useful additional information to assist readers to understand the 3D localization of the fibrils within the mitochondria and cells and would be a helpful addition.

Author Response 5

We thank for the comment. We have tried to obtain the 3D images for the **Mito-FF**. However, a lot of noise limit us to obtain a clear 3D image for **Mito-FF** because pyrene showed poor blue fluorescence in the most of confocal microscope. Meanwhile, **Mito-FF-NBD** showed bright green fluorescence above 40 μM , and is convenient for high resolution images. Thus, we have collected 3D images to understand co-localization of **Mito-FF-NBD** inside mitochondria. As shown in Supplementary Fig. 14, 6 slices were collected and **Mito-FF-NBD** showed a good overlap with MitoTracker with $R_r +0.8$.

Supplementary Figure 14 | a) 2D image showing co localization of **Mito-FF-NBD** inside mitochondria b) 3D localization for **Mito-FF-NBD** (middle: zoom view, right: orthogonal view). Inset: Localization. Blue: Nuclei stain, Red: MitoTracker Red FM, Green: **Mito-FF-NBD** (left) Zoom image (middle) Orthogonal view (right).

Comment 6.

The Mito-FF fibrils are suggested penetrate the membrane, as they appear on the outside of the cell membrane in image 4p of mouse cells. It is very difficult to see any evidence of these fibrils within these cells or across the membrane and an alternative hypothesis could be that in this instance the fibrils had assembled on the outside of the cell. I suggest more moderate language be used to describe the possibility of the fibrils penetrating as this is not really direct evidence. A time series or image that showed fibrils either side might be more conclusive.

Author Response 6

We agree with the reviewer's comments that the TEM is not a direct evidence to claim the formation of fibrils inside the mitochondria. As the reviewer suggests, we have changed the description in the manuscript with a moderate language in the page 10. Our series of data presented in the manuscript supports the formation of fibrils happens in the mitochondria. During our confocal experiments, we observed that the dysfunction of mitochondria of the HeLa cell after treatment with **Mito-FF** occurs quite fast within 3 h (Fig. R2). Therefore, it was very hard to get good TEM images by time dependent experiments. We observed that most of mitochondria are destroyed and a few mitochondria could be observed with fiber penetration.

Figure R2 |. TEM images of mitochondria within the HeLa cell showing the morphological damage induced by **Mito-FF** fibrils. TEM images were taken after 3 h incubation of **Mito-FF** with isolated mitochondria.

Revised version in the manuscript, page 11

A TEM experiment for HeLa cells treated with **Mito-FF** (20 μ M for 3 h) was conducted to investigate the structural change of mitochondria induced by intra-mitochondrial fibril formation. As shown in Fig. 4p, the mitochondria were found to be severely damaged with distorted membrane after **Mito-FF** treatment, and there were devoid of normal mitochondria morphology within HeLa cells. It was observed that the fibrils were centered within the destroyed mitochondria. It might be expected that fibril formation inside the mitochondria induces damage to the mitochondria with membrane disruption. However, the mitochondria in HeLa cells treated with **Mito-GG** remained unaffected (Fig. 4q) under similar conditions and appeared identical to the control (Fig. 4r). These results indicate that fibril assembly plays an important role in mitochondria dysfunction by disruption of the mitochondrial membrane, whereas spherical assembly does not.

Comment 7.

The high positive charge and hydrophobicity of the Mito-FF surface are described as being key reasons behind the proposed mechanism. Have these materials properties been described previously? If so can a reference be provided? Alternatively can the characterization data be presented?

Author Response 7

We propose that **Mito-FFs** accumulate and induce fibril formation within the cancer cell mitochondria triggered by the high membrane potential. The hydrophobicity helps to accumulate inside mitochondria, and the high positive charge of **Mito-FF** favors their interaction with the membrane as well. The material property of **Mito-FF** molecules has not characterized in any other previous reports. To get insight about the property of **Mito-FF**, we have measured the surface charge above their CAC. Our analysis showed that **Mito-FF** possess a high surface charge of $+43 \pm 1.3$ mV above CAC as measured with Malvern Zetasizer. However, **Mito-GG**, the micelle forming peptide, showed lower surface charge of $+23 \pm 1.9$ mV. We have included this information in the Supplementary Fig. 21. The hydrophobicity measurement measured via octanol partitioning by a modification of the shake-flask method, showed that **Mito-FF** possess high lipophilicity with a log P value of -0.075 compared with **Mito-GG** (-0.23). Based on these results, we propose that the fibrils with high positive charge and hydrophobicity induce mitochondria membrane disruption. Several reports suggest that fibrils could effectively induce liposomal membrane disruption. For example, Wells *et al.* described that the chemical fibrils formed by '1541' partially localize inside lysosome and induces lysosomal leakage, consequently initiate apoptosis cascade to cell death (Wells *et al. Nat. Chem. Biol.* **10**, 969-976 (2014), we added this on reference #22). In addition, it is well-known that positive charged nanostructures can induce endosomal escape by interaction between positive surface and negative membrane during endocytosis of nanostructure in the gene delivery research field (Park *et al. J. Pharm. Pharmacol.* **55**, 721-734 (2003), we added this on reference #23). Considering these reported literatures, we proposed the mechanism of membrane disruption by fibrils which has highly positive charge and rigid fiber morphology.

Revised version in the manuscript, page 11

The high positive surface charge ($+43 \pm 1.3$ mV, Supplementary Fig. 21) and hydrophobicity of **Mito-FF** fibrils favors strong interactions with the mitochondrial membrane, which could destroy the negatively charged mitochondrial membrane^{22,23}. To determine whether the fibrous assembly could induce membrane disruption, we conducted a dye-leakage assay using a model liposome encapsulated with a self-quenching dye (calcein).

Supplementary Figure 21 | Surface change analysis of a) **Mito-FF** b) Mito-GG in water above their CAC showing a surface charge of $+43\pm 1.3$ mV for **Mito-FF** nano fiber and $+23\pm 1.9$ for micelle respectively.

Comment 8.

Propidium iodide is shown to be excluded from the cells treated by Mito-FF in Figure 6g. As there is very little staining in this image it would be good to include a control image that shows positive PI staining using say an apoptotic chemical and a later stage of apoptosis.

Author Response 8

We agree with the reviewer's comment. According to our observation, cells were entered into an early apoptotic stage after treating with **Mito-FF** which was indicated by the appearance of green fluorescence with the treatment of Annexin V-FITC. However, cells remind impermeable to PI suggesting absence of necrosis or late apoptosis. To visualize the positive PI staining we have treated the HeLa cells with 500 μ M of H₂O₂ to induce necrosis (-Annexin, +PI) and included as the Supplementary Figure in comparison with **Mito-FF** (+Annexin, -PI), untreated control (-Annexin, -PI) and late apoptotic cells (+Annexin, +PI) induced by camptothecin(CPT) drug (10 μ M) according to the reported protocol.

Revised manuscript, page 16.

Propidium iodide (PI) and FITC-annexin V staining assays showed annexin V staining on the plasma membrane of **Mito-FF**-treated HeLa cells, but PI was excluded from the cells (Fig. 6g), suggesting that the cells entered in an early apoptotic stage within 6 h of treatment with **Mito-FF**²⁶. H₂O₂ (500 μ M) was used as the positive necrotic control which showed red staining of PI as a result of necrosis and Camptothecin (20 μ M) was used as a positive control for late apoptosis (stains both PI and annexin V). The membrane impermeable PI discriminates live or early apoptotic cells from late apoptotic or necrotic cells that lose membrane integrity. Annexin V stains both apoptotic cells, which expose phosphatidylserine extracellularly, and necrotic cells, which lose membrane integrity. (Supplementary Fig. 25). Moreover, the cellular morphology with membrane blebbing after treating with **Mito-FF** clearly indicates that cells undergo apoptosis (Supplementary Fig. 26) Fluorescence-activated cell sorting (FACS) was conducted to quantitatively analyze the apoptosis at successive time points (2 h, 4 h and 10 h) with the FITC-annexin V/PI staining assay (Fig. 6h, Supplementary Fig. 27.)

Supplementary Figure 25 | Confocal images with Annexin/PI staining on HeLa cells. Images show no staining by Annexin V and PI in the live cells (first row). +Annexin V, -PI staining after treating with **Mito-FF** (second row). -Annexin, +PI after inducing necrosis by H₂O₂ (third row) and +Annexin, +PI after inducing late apoptosis by Camptothecin (10 μM) (fourth row) in the HeLa cells for an incubation period of 12 h.

Comment 9.

The references to supplementary Figure 18, 19 and 20 appear to be missing from the main text. Without these references or discussion there doesn't seem much point in including these Figures.

Author Response 9

Thank you for the comment. These indicated that **Mito-VV** and **Mito-F_xF_x** also showed similar results with **Mito-FF** in the cellular death mechanism. We agree with the reviewer's comment. We think removing these figures are better since it is just a supporting data, so we have removed these figures from the original supplementary information.

Comment 10.

Supplementary Figure 22 indicates that the peptides accumulate in cancerous cells but to a lower extent in normal cells. Why would this be the case? Most of this work has also been produced on cell lines. Were primary isolates examined? The work is promising but have the authors considered the next steps in the evaluation of their OLISA approach? Once the cell undergoes apoptosis what are the likely clearance rates for these molecules and persistence in the body? Is anything known for these molecules or for molecules like those of this study? A comment would be worthwhile.

Author Response 10

This is important and valuable comment to us. We have analyzed the cellular toxicity in both cancer and normal cells. Our results indicated that **Mito-FF** was less toxic to the normal cells. We hypothesized that lower accumulation of **Mito-FF** inside the normal cells might be the reason for the lower toxicity. To confirm this, we have checked the mitochondrial accumulation in the normal cells. It showed lower accumulation of **Mito-FF** in the normal HEK293T cells than cancer (HeLa) as shown in Fig. 3a and Supplementary Fig. 28(Supplementary Fig. 22 in original version). The reason for the selectivity of **Mito-FF** towards cancer cells is related with the difference in the membrane potential of cancer and the normal, as previous reports suggests the mitochondrial membrane of the cancer cells are hyperpolarized (-180 to -220 mV) than the normal cells are (-120 to -160 mV) (*PloS one* **2013**, 8, e54346, *Annu. Rev. Pharmacol. Toxicol.* **2007**, 47, 629 – 656, *Adv. Drug Delivery Rev.* **2001**, 49, 63 – 70). The difference of about 60 mV in the membrane potential could results in tenfold increase in uptake. Thus, we concluded from our experimental results that the **Mito-FF** accumulated to a lower extent inside the normal cell mitochondria due to the membrane potential difference.

As reviewer points out, in this work, we have focused on the impact of OLISA to control the cell fate *in vitro*. As the next step of our study, we will investigate the cellular toxicity using primary isolated cell, *in vivo* metabolic stability, pharmacokinetics of our compounds with a xenograft tumor model. We think that even though the membrane potential difference between the normal and cancer could bring the effective selectivity in *in vitro* studies, *in vivo* required even more specificity to cancer (i.e. conjugation of **Mito-FF** with cancer targeting moiety like RGD, or folic acid). In addition, the peptide might be easily degraded in the body due to many proteolytic enzyme. If the peptides reach to cancer and induces apoptosis of cancer before degradation, it will have high potential for cancer therapy, because it will be cleared by enzyme. If the peptides are degraded quickly before reaching cancer cells, then, it is necessary to investigate the impact of D-Phe analogue of **Mito-FF** to get more insight about the *in vivo* stability since D-amino acids are not well recognized by natural enzyme. We will come up with those results of OLISA such as *in vivo* efficacy, clearance rate, selectivity etc. in the near future.

Minor points:

Comment 11.

In Figure 2 c there are a number of bands assigned and highlighted within the spectra but only one is discussed in the text. Are the other bands significant? If so can they be described?

Author Response 11

The band near 1735 cm⁻¹ (Amide I characteristic band) is associated with the C=O stretching vibration and directly related to the backbone conformation. The band at 1675 cm⁻¹ indicates the presence of β -turn confirmation. However, in the manuscript we just would like to mention the presence of β -confirmation in the **Mito-FF**, because an in-depth investigation of dipeptide was previously reported. In the present manuscript, it might be not significant, so we have excluded the description about other two bands.

Comment 12.

On page 4 – the description of ‘similar uptake’ at 4°C is confusing. Relative to what? Perhaps this just needs rephrasing.

Author Response 12

We agree with the reviewer’s comment. We have rephrased the sentence as “Cellular uptake analysis at 4°C showed similar uptake as observed at 37 °C, suggesting that the peptide enters via an energy-independent pathway (Supplementary Fig.7)”.

Comment 13.

On page 4 Mito-FF is described as diffusing through the plasma membrane. Is there any prior information on the diffusion of FF structures through the membrane? A reference would be useful here.

Author Response 13

From the literature survey we concluded that the FF containing short peptides in the molecular state could easily diffuse through the plasma membrane without the assistance of any endocytosis mechanism, which supports our experimental evidence of the cellular entry of **Mito-FF** at 4 °C (*J. Am. Chem. Soc.* **137**, 10040-10043 (2015)). In contrast, the assembled structure (e.g. fibrils) usually requires energy dependent pathway for the cellular entry. We would like to mention that the energy independent, diffusion based cellular uptake of FF containing structure is previously explained in the reference #13. We have included citation for this reference in the manuscript.

Comment 14.

On page 6 – significant figures appear to be overstated e.g. 10.331 mM internal cell Mito-FF concentration.

Author Response 14

We agree with the comment and we have changed it in the manuscript as 10.3.

“Our calculations showed that the mitochondrial accumulation of **Mito-FF** increased in accordance with the original concentration in the culture medium, i.e., 2.7 mM and 10.3 mM for external concentrations of 5 μ M or 10 μ M, respectively, suggesting that the self-assembly of **Mito-FF** inside the mitochondria could readily occur as a result of the high local concentration (Fig. 3a).”

Comment 15.

It is worthy of comment in the main text that the intensity of staining in supplementary Figure 11 is quite low – i.e. there is only limited evidence of ROS production via this method.

Author Response 15

We apologize for the poor resolution due to low intensity of emission for the images. Now we changed high resolution images to show ROS production inside mitochondria monitored with MitoSOX Red which were presented in Supplementary Fig. 17 and Fig. 19. **Mito-FF** induces considerable ROS within mitochondria as indicated by the following images of Mito-SOX (5 μ M working concentration) and the mitochondria where severely damaged after treating with the **Mito-FF**.

Supplementary Figure 17 | ROS generation inside the mitochondria monitored by the red fluorescence from MitoSOX. **Mito-FF** was incubated for 6 h with HeLa cells and stained with MitoSOX before 10 minute of measurement and analyzed via confocal microscopy. The red fluorescence form MitoSOX confirmed the ROS inside mitochondria.

Supplementary Figure 19 | ROS generation inside the mitochondria monitored by the red fluorescence from MitoSOX. **Mito-VV** or **Mito-F_xF_x** was incubated for 6 h with HeLa cells and stained with MitoSOX before 10 minute of measurement and analyzed via confocal microscopy. The red fluorescence form MitoSOX confirmed the ROS inside mitochondria.

Comment 16.

Supplementary Figure 21- the labelling of the dye on each axes for the FACS appears missing from the legend.

Author Response 16

We have replaced the figure legend in the supplementary information as follows:

Supplementary Figure 27 | Flow cytometric analysis of **Mito-FF** were obtained with Annexin/PI staining on HeLa cells after treating with **Mito-FF** for desired time point. Control experiment was performed without **Mito-FF**.

REVIEWERS' COMMENTS:

Reviewer #1 (Remarks to the Author):

The authors have addressed my comments and the paper can be published.

Reviewer #2 (Remarks to the Author):

The authors have provided substantial further evidence to support their manuscript and have addressed the comments well. I am satisfied with the additional data and clarifications presented.